## RESEARCH ARTICLE

# Glutamate decreases oxidative stress and lipid droplet formation in astrocytes

Luis Fernando Rubio-Atonal[1,2,3], Jinlan Chang[1], Julie Jacquemyn[1,2], Isha Ralhan[1,2], Iset Ilarraza[1] and Maria S. Ioannou[1,2,3,4,5],*

## ABSTRACT

Astrocytes degrade fatty acids in response to glutamate while reducing the abundance of lipid droplets. But how glutamate regulates lipid droplets in astrocytes is unclear. Here, we used primary rat astrocytes to show that glutamate decreases the amount of reactive oxygen species, which, in turn, reduces autophagy and the amount of lipids in need of storage in lipid droplets. This decrease in lipid droplets and reactive oxygen species occurs independently of glutamate import through excitatory amino acid transporters (EAATs). However, activation of AMPK, downstream of EAATs, further promotes a decrease in lipid droplets. Glutamate also increases the pool of fused mitochondria capable of maintaining enhanced fatty acid metabolism. Our work reveals how astrocytic metabolism is regulated by glutamate, which can serve to coordinate astrocyte physiology with neuronal activity.

KEY WORDS: Fatty acid oxidation, Lipid droplet, Autophagy, Oxidative stress, Lipid peroxidation, Astrocytes

## INTRODUCTION

The metabolic coupling between neurons and astrocytes is crucial for neural physiology in health and disease (Bonvento and Bolaños, 2021). Impaired energy metabolism is a converging feature of neurodegenerative disease and is often associated with defects in the coordination between these two cell types (Bonvento and Bolaños, 2021). Yet, much remains unknown regarding how this coordination is regulated, particularly when it comes to its influence on lipid metabolism.

A key mechanism of metabolic coupling is the transfer of lipids from neurons to astrocytic lipid droplets (Ioannou et al., 2019a; Liu et al., 2015). This pathway is especially prominent during oxidative stress, including excitotoxicity, but also occurs under physiological conditions, such as during the sleep-wake cycle (Haynes et al., 2024; Ioannou et al., 2019a,b). This helps protect neurons from a toxic accumulation of lipids and relies on astrocytes to safely store them as neutral lipids. Neurons also depend on astrocytes to degrade fatty acids, a major component of most lipids, by mitochondrial

β-oxidation (Mi et al., 2023; Qi et al., 2021). Although astrocytes primarily rely on glycolysis to meet their energy demands, their mitochondrial proteome is enriched in enzymes responsible for β-oxidation of fatty acids (Fecher et al., 2019). When astrocytic fatty acid oxidation is impaired, neurons become vulnerable to oxidative stress and degenerate (Mi et al., 2023).

Given the impact of astrocytic lipid storage and catabolism on neuronal health, it is perhaps not surprising that neurons can influence this process. For example, glutamate is the major excitatory neurotransmitter in the brain. Glutamate stimulates transport of fatty acids into the mitochondria, where fatty acid oxidation occurs and reduces the abundance of lipid droplets in astrocytes (Ioannou et al., 2019a). Whether the reduction in lipid storage depends on fatty acid oxidation is unclear. In addition, astrocytes express excitatory amino acid transporter 1 and 2 (EAAT1/2; also known as SLC1A3/2), which import glutamate from the extracellular space (Todd and Hardingham, 2020). Once inside the cell, glutamate can regulate signaling pathways known to influence fatty acid oxidation (Boone et al., 2000). Whether glutamate import is required for alterations in lipid storage is unknown.

Here, we demonstrate that glutamate reduces lipid droplets in astrocytes independently of glutamate uptake or changes in fatty acid oxidation. Instead, glutamate reduces the amount of reactive oxygen species (ROS). This, in turn, reduces autophagy, thereby decreasing the amount of lipids in need of storage in lipid droplets. Additionally, this leads to an increased pool of fused mitochondria capable of maintaining increased oxidative metabolism. Collectively, this work reveals how astrocytic metabolism is regulated by glutamate, which likely serves to coordinate astrocytic physiology with neuronal activity.

## RESULTS

### Astrocytes oxidize exogenous fatty acids in response to glutamate

We previously found that glutamate modulates lipid physiology in astrocytes grown in serum (Ioannou et al., 2019a), which is known to induce a reactive phenotype typically associated with disease (Liddelow et al., 2017). To explore the effects of glutamate on lipids in resting-state astrocytes, we cultured primary rat astrocytes in a serum-free medium that maintains astrocytes in a non-reactive state (Foo et al., 2011). These astrocytes had a more ramified morphology, relative to the fibroblast-like appearance of those grown in serum (Fig. 1A), and ~95% of cells stained positive for the astrocyte marker GFAP (Fig. 1B). Because this culture condition alters the physiology of astrocytes compared to that of those grown in serum (Prah et al., 2019), we tested whether these astrocytes similarly respond to glutamate by increasing fatty acid transport into the mitochondria (Ioannou et al., 2019a). Here, astrocytes were loaded with the fluorescently tagged saturated fatty acid BODIPY 558/568 C12 (Red-C12), and the labeled fatty acid was tracked in the presence

[1]Department of Physiology, University of Alberta, Edmonton, AB T6G 2H7, Canada. [2]Group on Molecular and Cell Biology of Lipids, University of Alberta, Edmonton, AB T6G 2H7, Canada. [3]Women and Children's Health Research Institute, University of Alberta, Edmonton, AB T6G 2H7, Canada. [4]Department of Cell Biology, University of Alberta, Edmonton, AB T6G 2H7, Canada. [5]Neuroscience and Mental Health Institute, University of Alberta, Edmonton, AB T6G 2H7, Canada.

*Author for correspondence (ioannou@ualberta.ca)

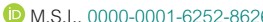 M.S.I., 0000-0001-6252-8626

Journal of Cell Science

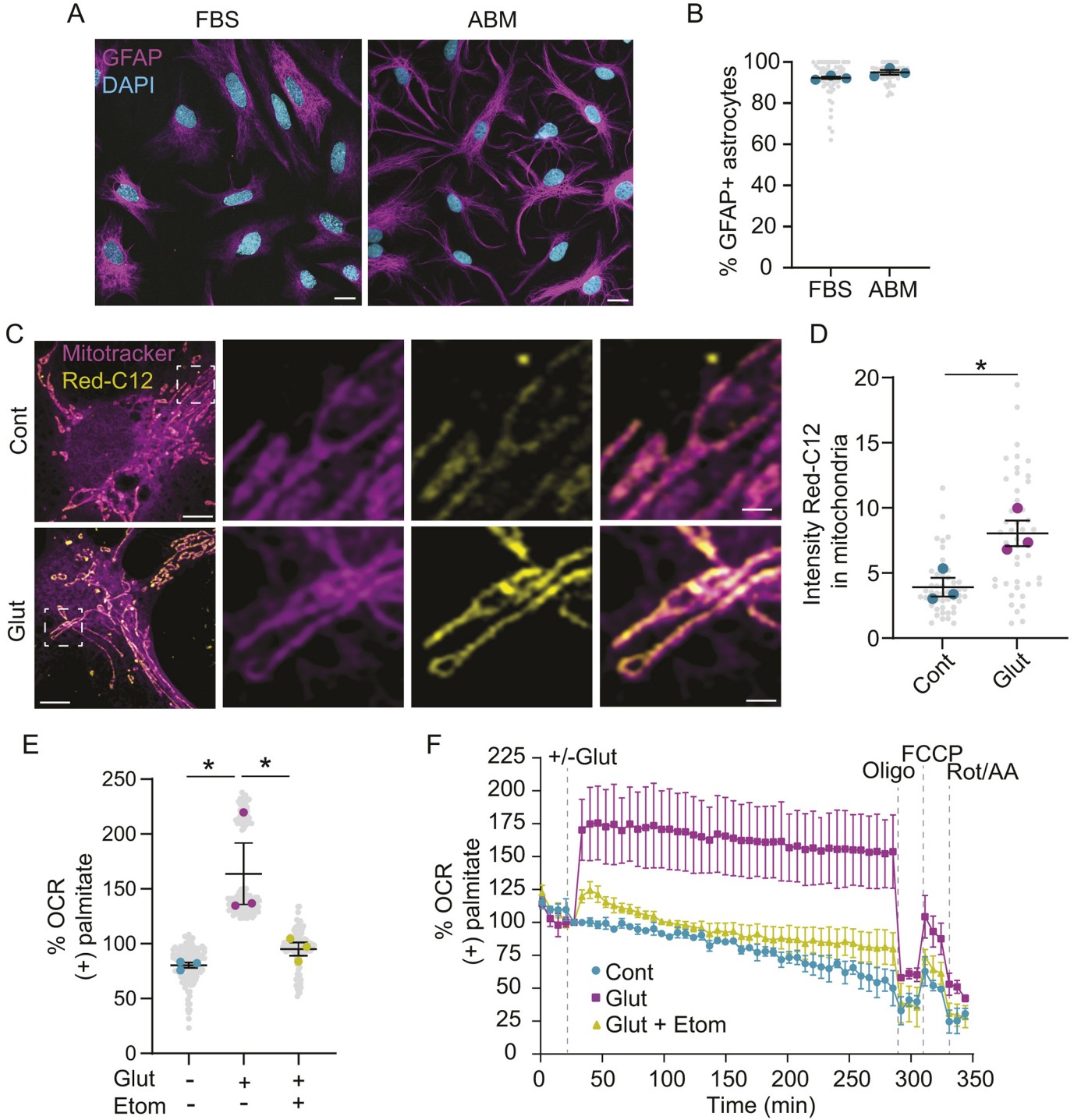

**Fig. 1. Glutamate increases mitochondrial oxidation of exogenous fatty acids.** (A) Representative images of astrocytes cultured in fetal bovine serum (FBS) or astrocyte basal medium (ABM) and stained for GFAP. Scale bars: 10 μm. (B) Percentage of DAPI-positive cells also positive for GFAP for FBS- and ABM-cultured astrocytes. Mean±s.e.m. *n*=3 independent experiments, ten technical replicates per *n*. (C) Representative images of astrocytes loaded with Red-C12±glutamate (Glut) and stained with MitoTracker Deep Red. Scale bars: 5 μm for images on left and 1 μm for boxed area magnified on right. (D) Quantification of relative intensity of Red-C12 in MitoTracker Deep Red-positive mitochondria. Mean±s.e.m. *n*=3 independent experiments, 15 technical replicates per *n*. Unpaired two-tailed *t*-test, *$P<0.05$. (E) Oxygen consumption rate (OCR) of astrocytes supplied with palmitate-BSA±glutamate and etomoxir (Etom). Mean±s.e.m. *n*=3 independent experiments, 40 technical replicates per *n*. One-way ANOVA with Dunnett's post-test, *$P<0.05$. (F) OCR of astrocytes supplied with palmitate-BSA±glutamate and etomoxir, and sequentially treated with oligomycin (oligo), FCCP, and rotenone and antimycin A (Rot/AA). Mean±s.e.m. *n*=3 independent experiments, four to 12 technical replicates per *n*.

or absence of glutamate. Given that extracellular glutamate concentration in the brain varies by several orders of magnitude depending on the compartment and/or biological context (Dzubay and Jahr, 1999; Kalivas, 2011), 100 μM glutamate was used, as this concentration was previously shown to increase lipid transport from neurons to astrocytes and reduce astrocyte lipid droplets (Ioannou et al., 2019a). Consistent with previous work, glutamate increased the amount of Red-C12 trafficked to mitochondria labeled with

MitoTracker Deep Red (Fig. 1C,D). Because fatty acids are catabolized in the mitochondria, we next performed a metabolic flux assay to determine whether glutamate increased fatty acid oxidation in the presence of exogenous palmitate. Indeed, we observed an increase in oxygen consumption rate upon glutamate treatment, which was blocked by etomoxir, a carnitine palmitoyltransferase-1 inhibitor that prevents fatty acid transport into mitochondria (Fig. 1E,F). These data demonstrate that serum-free astrocytes oxidize exogenous fatty acids in response to glutamate treatment.

We next investigated how glutamate affects astrocyte metabolism in the absence of exogenous fatty acids. Without exogenous palmitate, astrocytes increased oxygen consumption rate and mitochondrial ATP generation in response to glutamate (Fig. 2A–C). However, this increased oxidative metabolism was unaffected by etomoxir, suggesting that it is independent of fatty acids (Fig. 2A–C). To determine what substrate astrocytes were using for oxidative metabolism in these conditions, we performed metabolic flux assays using inhibitors of different fuel pathways. These assays confirmed

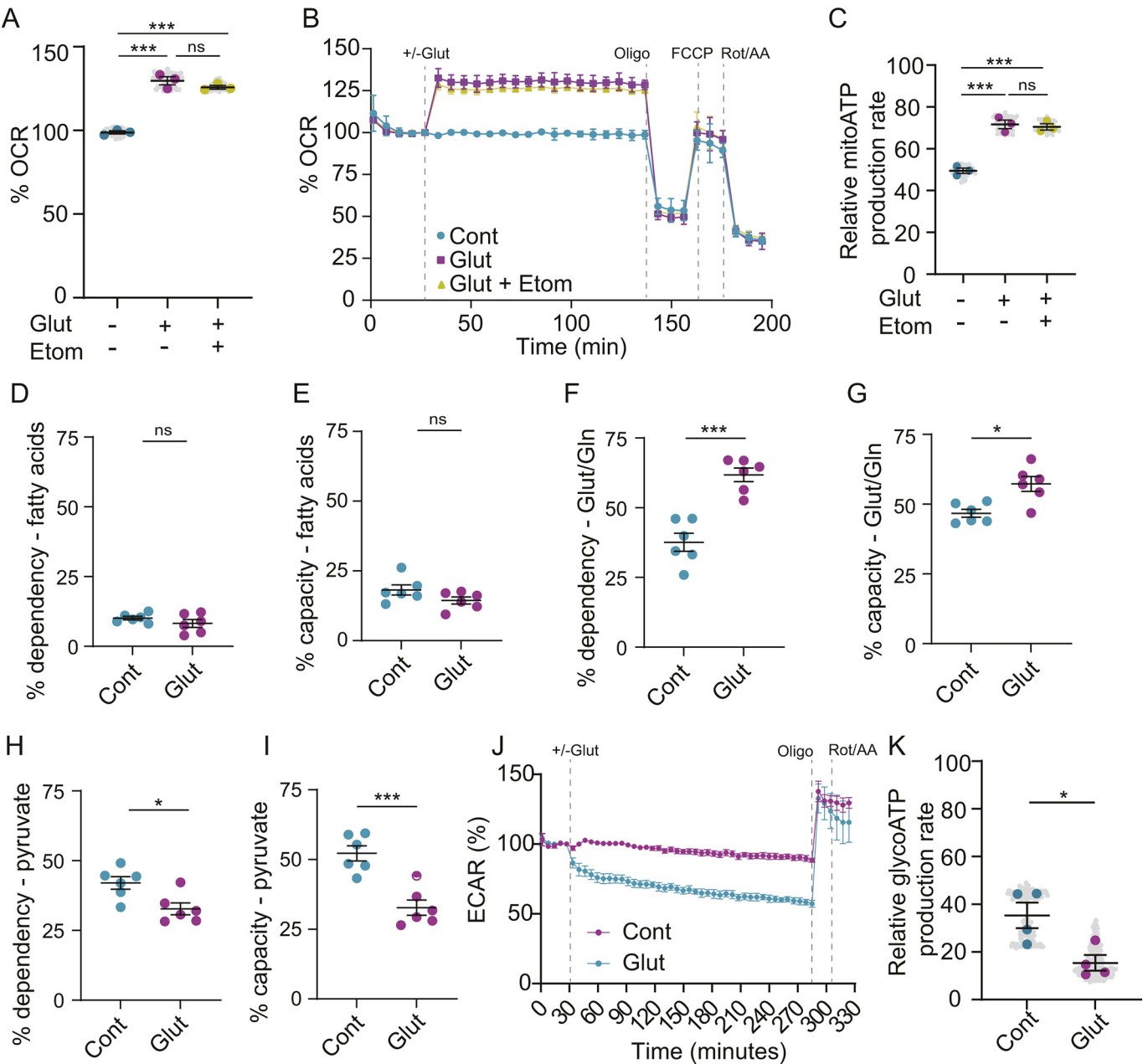

**Fig. 2. Fatty acid oxidation is unaffected by glutamate in the absence of exogenous fatty acids.** (A,B) OCR of astrocytes±glutamate and etomoxir (A) and of those also sequentially treated with oligomycin (oligo), FCCP, and rotenone and antimycin A (B). Mean±s.e.m. $n$=3 independent experiments, 17 technical replicates per $n$. One-way ANOVA with Tukey's post-test, ***$P$<0.001. (C) Rate of mitochondrial ATP production of astrocytes±glutamate and etomoxir. Mean±s.e.m. $n$=3 independent experiments, 17 technical replicates per $n$. One-way ANOVA with Tukey's post-test, ***$P$<0.001. (D–I) Dependency (D,F,H) and capacity (E,G,I) of astrocytes to use different fuel types: fatty acids (D,E), glutamate/glutamine (Glut/Gln; F,G) and pyruvate (H,I). Mean±s.e.m. $n$=6 independent experiments, calculations were performed on the averages from eight technical replicates for each $n$. Unpaired two-tailed $t$-tests, *$P$<0.05, ***$P$<0.001. (J–K) Extracellular acidification rate (ECAR) of astrocytes±glutamate. Mean±s.e.m., $n$=4 independent experiments, 40 technical replicates per $n$. Unpaired two-tailed $t$-tests, *$P$<0.05. ns, non-significant.

that the dependence and capacity of astrocytes to oxidize fatty acids was unaffected by glutamate (Fig. 2D,E). Instead, astrocytes increased their utilization of glutamate/glutamine (Fig. 2F,G), while decreasing their use of pyruvate (Fig. 2H,I), upon glutamate treatment. This is consistent with reports that glutamate oxidation spares astrocytes the need to catabolize glucose (McKenna, 2013). Consistently, glutamate decreased glycolysis as indicated by reduced extracellular acidification rates and glycolytic ATP production (Fig. 2J,K). Collectively, these results support a shift to oxidative metabolism in response to glutamate, which includes fatty acid oxidation when extracellular lipids are abundant.

### Neutral lipolysis is unaffected by glutamate

Because glutamate decreases the number of lipid droplets in astrocytes grown in serum (Ioannou et al., 2019a), we tested whether the same was true for serum-free astrocyte cultures. Indeed, glutamate reduced the number of lipid droplets in serum-free growth media (Fig. 3A,B). Lipid droplet size, however, remained unaffected (Fig. 3C). The same effect was observed when astrocytes were co-treated with oleic acid to promote lipid droplet formation (Fig. 3D–F). Collectively, these data indicate that glutamate

reduces lipid droplets independently of changes in fatty acid oxidation and support the use of serum-free astrocyte cultures to explore the effects of glutamate on lipid droplet physiology.

We next wondered how glutamate reduces lipid droplets. Increased lipolysis, whereby lipases on the surface of lipid droplets degrade triglycerides into free fatty acids, could reduce the pool of lipid droplets. The brain uses DDHD2 as the major triglyceride lipase, which can be inhibited by the drug KLH45 (Inloes et al., 2014). We found that KLH45 rescued lipid droplet numbers back to control levels in the presence of glutamate (Fig. 4A,B). This confirmed that astrocytes use DDHD2 to break down triglycerides. However, alterations in the rates of lipolysis typically change the size of lipid droplets. Stimulating lipolysis shrinks lipid droplets, whereas inhibiting lipolysis causes lipid droplets to expand in size (Inloes et al., 2014; Schott et al., 2019). Whereas inhibiting DDHD2 with KLH45 caused lipid droplets to become larger (Fig. 4A,C), glutamate alone had no effect on lipid droplet size (Fig. 3A,C and Fig. 4A,C). We also reasoned that if lipolysis was increased, there would be an increase in free fatty acids and/or acylcarnitines in astrocytes. However, glutamate did not affect the levels of free fatty acids, while total acylcarnitines decreased (Fig. 4D–F).

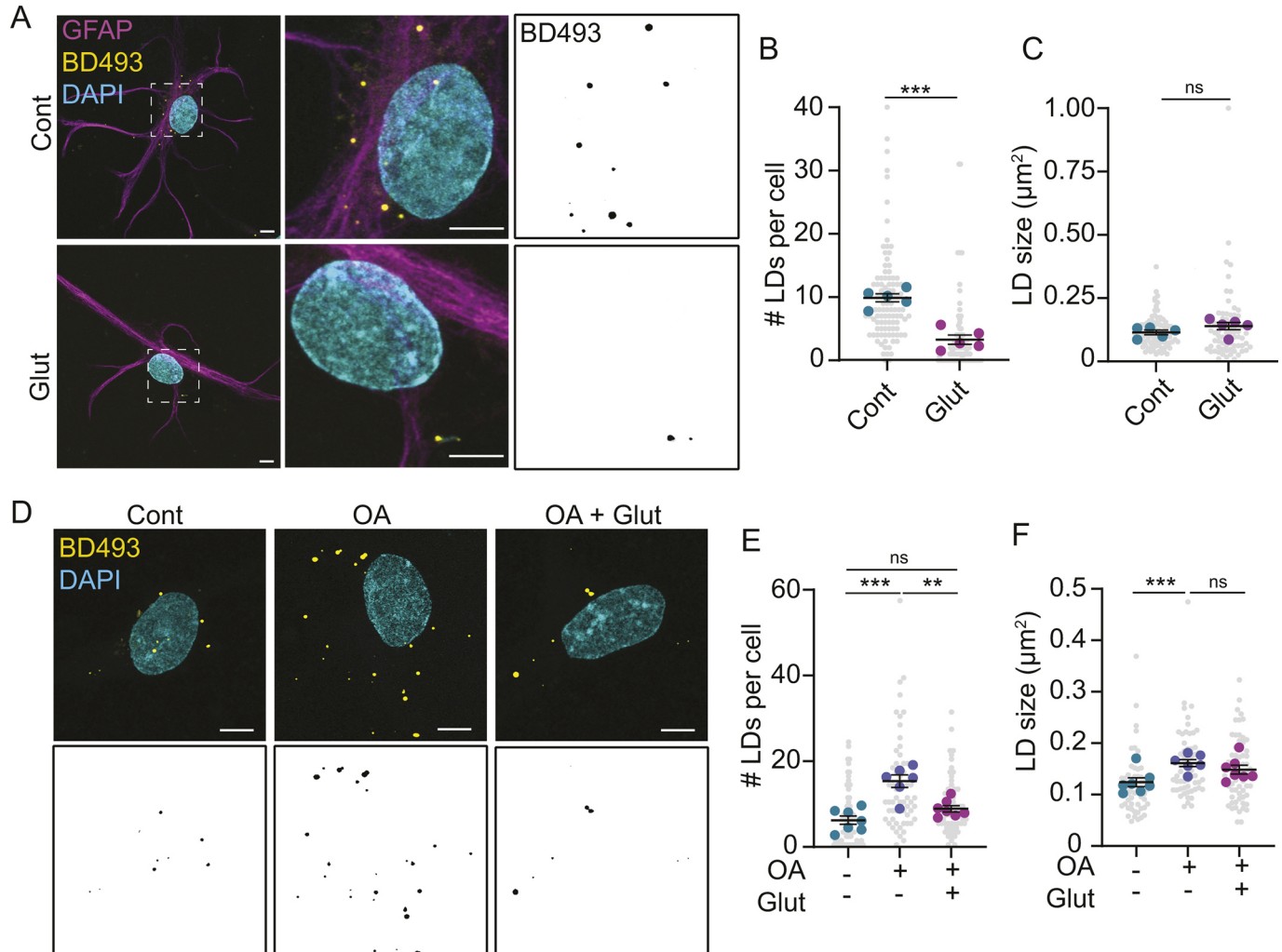

Fig. 3. Glutamate reduces astrocytic lipid droplet numbers. (A–C) Astrocytes treated±glutamate in ABM. Cells were fixed, and lipid droplets (LDs) were stained with BODIPY 493/503 (BD493). Scale bars: 5 μm. Mean±s.e.m. $n$=5 independent experiments, 20 technical replicates per $n$ for B and nine to 20 technical replicates per $n$ for C. Unpaired two-tailed $t$-test, ***$P$<0.001. (D–F) Astrocytes treated±oleic acid (OA) and glutamate in ABM. Cells were fixed, and lipid droplets were stained with BD493. Scale bars: 5 μm. Mean±s.e.m. $n$=7 independent experiments, ten technical replicates per $n$. One-way ANOVA with Tukey post-test, **$P$<0.01, ***$P$<0.001. ns, non-significant.

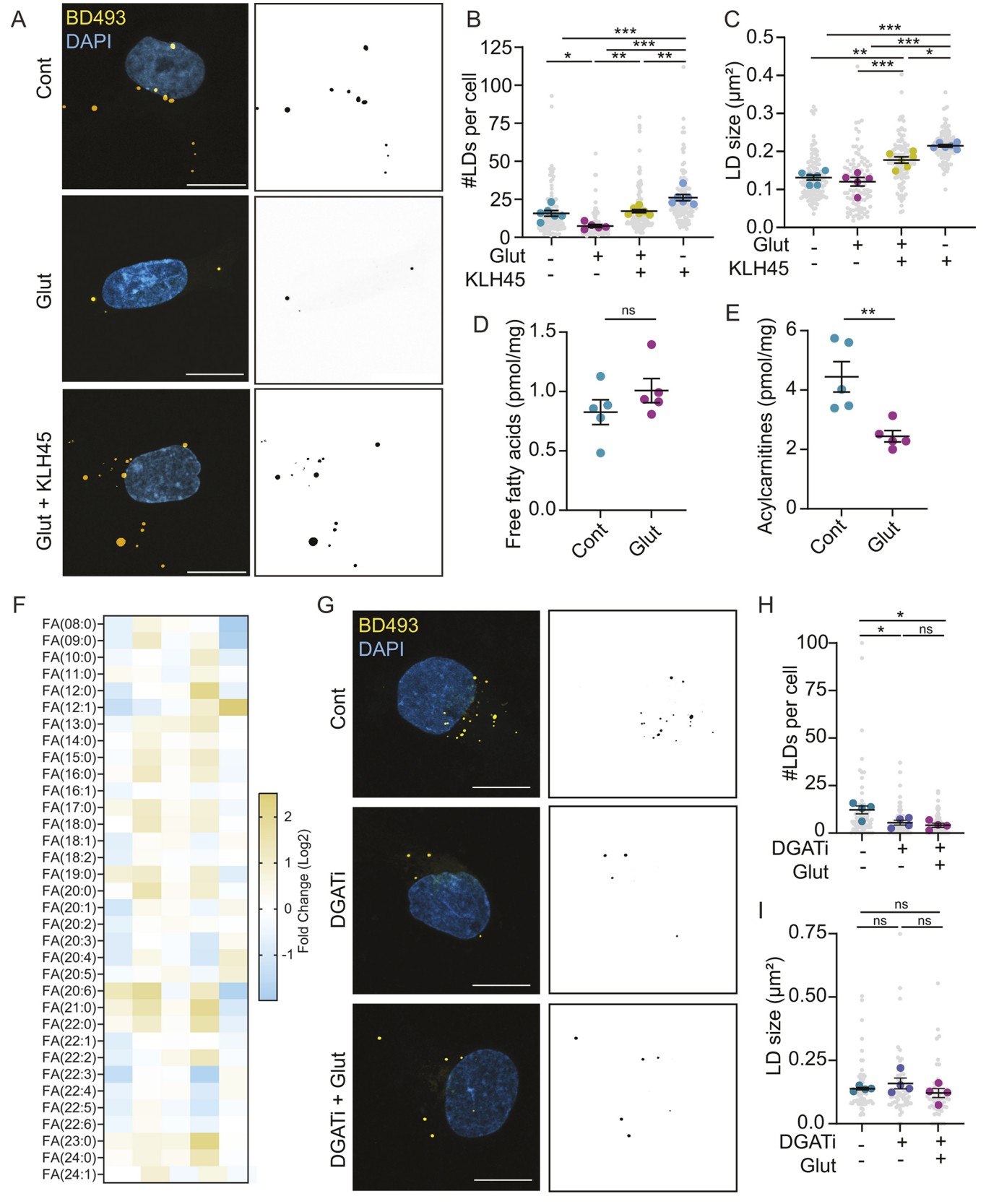

**Fig. 4.** See next page for legend.

Finally, we tested whether glutamate continues to reduce lipid droplets in the absence of triglyceride synthesis by inhibiting diacylglycerol acyltransferase (DGAT) 1 and 2. We reasoned that if lipolysis was increased, then glutamate would further reduce lipid droplets in the absence of DGAT1/2 activity. However, glutamate had no additional effect on lipid droplets in the presence of DGAT1/2

**Fig. 4. Steady-state lipolysis by DDHD2 is unaffected by glutamate.**
(A–C) Astrocytes treated±glutamate and KLH45 in ABM. Cells were fixed,
and lipid droplets were stained with BD493. Scale bars: 10 µm. Mean±s.e.m.
$n$=5-6 independent experiments, 20 technical replicates per $n$. One-way
ANOVA with Tukey post-test, *$P$<0.05, **$P$<0.01, ***$P$<0.001. Only
significant comparisons are shown. (D) Total free fatty acids in astrocytes
treated±glutamate. Mean±s.e.m. $n$=5 independent experiments. Unpaired
two-tailed $t$-test. (E) Total acylcarnitines in astrocytes treated±glutamate.
Mean±s.e.m. $n$=5 independent experiments. Unpaired two-tailed $t$-test,
**$P$<0.01. (F) Heat map showing Log2 fold change of fatty acid (FA) species
altered in glutamate-treated astrocytes compared to controls. $n$=5
independent experiments. Multiple $t$-test with Holm–Šídák correction.
No significant differences were observed. (G–I) Astrocytes treated±DGAT1
(PF-04620110) and DGAT2 (PF-06424439) inhibitors (DGATi) and
glutamate in ABM. Cells were fixed, and lipid droplets were stained with
BD493. Scale bars: 10 µm. Mean±s.e.m. $n$=4 independent experiments,
13-17 technical replicates per $n$. One-way ANOVA with Tukey post-test,
*$P$<0.05. ns, non-significant.

inhibitors (Fig. 4G–I). Collectively these data suggest that the steady
rate of lipolysis in astrocytes is unaffected by glutamate.

## Glutamate reduces astrocytic autophagy

Another mechanism to reduce lipid droplet abundance is lipophagy,
the autophagic degradation of lipid droplets (Singh et al., 2009). We
tested whether glutamate affects the lipidation of LC3B (also known
as MAP1LC3B), a key step in the biogenesis of autophagosomes.
These experiments were performed in the presence or absence
of bafilomycin A1, which inhibits late stages of autophagy by
disrupting lysosomal acidification and degradative function
(Mauvezin and Neufeld, 2015). Astrocytes showed decreased
lipidation of LC3B (denoted as LC3B-II) in the presence of
bafilomycin A1 (Fig. 5A,B). Glutamate also reduced autophagic
flux, calculated as the difference between LC3B-II in the presence
and absence of bafilomycin A1 (Fig. 5A,C). Similarly, we observed
a decrease in the number and area of LC3B-positive puncta
upon glutamate treatment in the presence of bafilomycin A1 by
microscopy (Fig. 5D–F). These data indicate that glutamate reduces
autophagy in astrocytes.

Lipophagy can also occur through the docking of lysosomes
onto lipid droplets and the direct transfer of neutral lipids into
the lysosomes (Schulze et al., 2020). To test this, we assessed
colocalization between BODIPY 493/503 (BD493)-positive lipid
droplets and acidic compartments stained with LysoTracker. We
found no difference in the percentage colocalization with glutamate
relative to total BD493-positive area, whereas the percentage
colocalization relative to total Lysotracker-positive area decreased
(Fig. 5G–I). Glutamate did not affect the number of Lysotracker-
positive puncta (Fig. 5J). This indicates that glutamate does not
increase the association of lipid droplets with lysosomes as would
be required for lipophagy. Collectively, these data suggest that
glutamate reduces autophagy, and, therefore, lipophagy is unlikely
to account for the reduction in lipid droplets observed. Rather, the
decreased autophagy might reduce the amount of lipids in need of
storage, resulting in fewer lipid droplets.

Next, we investigated whether mitochondrial morphology was
affected, as fused mitochondria are typically protected from
autophagic degradation (Rambold et al., 2011; Twig et al., 2008),
and we observed decreased autophagy. Indeed, the number of
TOMM20-positive mitochondria per cell was increased with no
change in total cell volume (Fig. 5K–M). Glutamate also increased
the average volume per mitochondrion, indicative of a more fused or
tubular network (Fig. 5N,O). This supports a role for glutamate in

maintaining a functional mitochondrial network capable of
enhanced oxidative metabolism.

## Glutamate activates AMPK signaling in astrocytes

These findings led us to wonder whether glutamate-induced
intracellular signaling was involved in regulating lipid droplets. We
first examined AMP-activated protein kinase (AMPK; also known as
PRKAB1), as glutamate activates AMPK in other cell types and
AMPK is an important regulator of autophagy. Indeed, glutamate
increased phosphorylation of AMPK on Thr172, indicative of
activation, while total AMPK levels remain unchanged (Fig. 6A–C).
We next assessed the activation of acetyl-CoA carboxylase 1/2 (ACC;
also known as ACACA/B), which occurs downstream of AMPK.
Phosphorylation of ACC by AMPK blocks ACC activity, resulting in
increased fatty acid transport into the mitochondria for β-oxidation
and decreased *de novo* fatty acid synthesis (Wang et al., 2022). We
found that phosphorylation of ACC on Ser79 was also increased
with glutamate treatment, while total ACC remained unchanged
(Fig. 6A,D,E).

To validate whether AMPK regulates lipid droplets in astrocytes,
we modulated AMPK activity. We found that the reduction in lipid
droplets by glutamate was prevented by co-treatment with compound
C, an AMPK inhibitor (Fig. 6F–H). Conversely, AMPK activators,
AICAR and metformin, reduced the number of lipid droplets in the
absence of glutamate (Fig. 6I–K). These data indicate that glutamate
activates AMPK in astrocytes, which can contribute to the reduction
in lipid droplets upon glutamate treatment.

## Glutamate reduces lipid droplets and ROS independently of internalization

Astrocytes can respond to extracellular glutamate via surface receptors
or uptake through glutamate transporters (Rose et al., 2018). We
sought to determine whether glutamate internalization is necessary for
the regulation of lipid droplets. Astrocytes primarily take up glutamate
from the extracellular space via EAAT1/2 (Rose et al., 2018). To
understand the effects of glutamate import, we blocked EAAT1/2
using the selective inhibitor (3S)-3-[[3-[[4-(trifluoromethyl) benzoyl]
amino] phenyl] methoxy]-L-aspartic acid (TFB-TBOA) (Fig. 7A).
Treatment with 200 nM TFB-TBOA had no effect on the reduction
in lipid droplet numbers caused by glutamate (Fig. 7B–D). This
indicated that the effects of glutamate on lipid droplets are
independent of glutamate internalization.

We previously found that glutamate also decreases ROS levels
in astrocytes (Ioannou et al., 2019a). Because glutamate can act
as a precursor to antioxidants such as glutathione (Hlozkova et al.,
2024), we wondered whether internalization of glutamate is
required for the protective effects observed. However, 200 nM
TFB-TBOA did not prevent the reduction in superoxide and
hydroxyl radicals by glutamate as indicated by CellRox staining
(Fig. 7E,F), and did not prevent the glutamate-induced reduction in
lipid hydroperoxides (Fig. 7G). These data indicate that glutamate
also increases the antioxidant capacity of astrocytes independently
of glutamate import.

Interestingly, inhibition of AMPK using compound C increased
lipid hydroperoxides in the presence of glutamate (Fig. 7G),
suggesting that AMPK signaling could regulate the antioxidant
effects of glutamate. As such, we tested whether the activation
of AMPK is also independent of glutamate uptake. However,
TFB-TBOA prevented the activation of both AMPK and ACC by
glutamate, with no changes in total levels of protein (Fig. 7H–L).
Collectively, these data indicate that AMPK activation is sufficient,
but not necessary, for the effects of glutamate on lipid droplets and

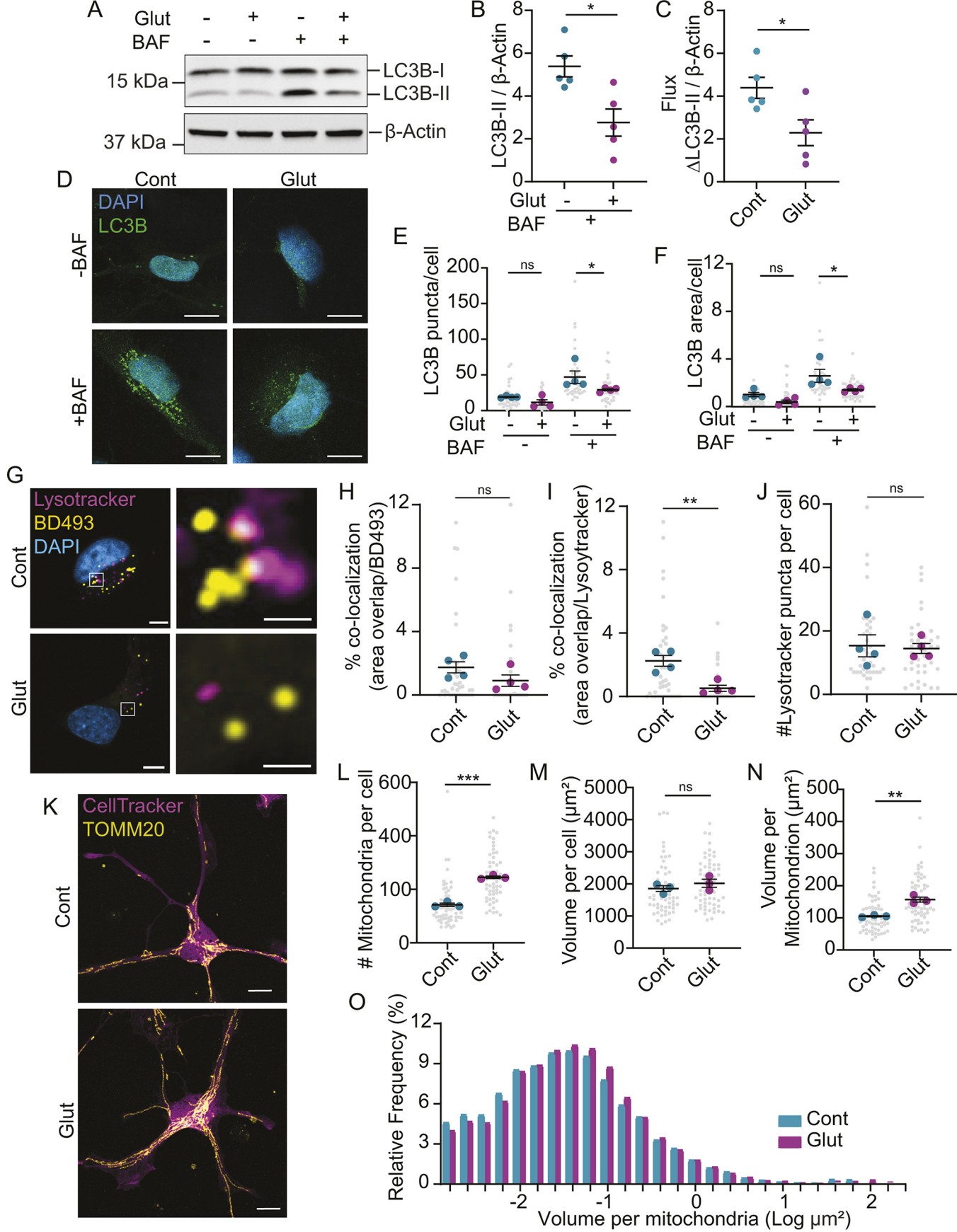

**Fig. 5.** See next page for legend.

**Fig. 5. Glutamate reduces autophagy and increases mitochondrial volume.** (A) Astrocytes treated±glutamate and bafilomycin A1 (BAF), with lysates analyzed for LC3B and actin by western blotting. Representative image for data shown in B and C. (B) LC3B-II normalized to β-actin in the presence of BAF. Mean±s.e.m. $n$=5 biological replicates from three independent experiments. Unpaired two-tailed $t$-test, *$P$<0.05. (C) Autophagic flux calculated as the difference in LC3B-II in the presence and absence of BAF. Mean±s.e.m., $n$=5 biological replicates from three independent experiments. Unpaired two-tailed $t$-test, *$P$<0.05. (D) Astrocytes treated±glutamate and bafilomycin A1, fixed and immunostained for LC3B. Scale bars: 10 μm. (E,F) The number (E) and area (F) of LC3B-positive puncta per cell were quantified. Mean±s.e.m. $n$=4 independent experiments, eight to ten technical replicates per $n$. One-way ANOVA with Šídák's post-test, *$P$<0.05. (G) Astrocytes treated±glutamate in ABM, stained with Lysotracker Red and BD493. Scale bars: 5 μm for left column and 1 μm for right column. Representative images for data shown in H–J. (H) Area overlap of BD493 and Lysotracker relative to total BD493 area. (I) Area overlap of BD493 and Lysotracker relative to total Lysotracker area. (J) Number of Lysotracker-positive puncta per cell. (H–J) Mean±s.e.m., $n$=4 independent experiments, ten to 11 technical replicates per $n$. Unpaired two-tailed $t$-test, **$P$<0.01. (K) Astrocytes treated±glutamate, stained with CellTracker Green and immunostained for TOMM20. Scale bars: 10 μm. Representative images for data shown in L–N. (L–N) Quantification of number or volume of mitochondria or cells. Mean±s.e.m. $n$=3 independent experiments, 20 technical replicates per $n$. Unpaired two-tailed $t$-test, **$P$<0.01, ***$P$<0.001. (O) Size distribution of mitochondria±glutamate treatment. $n$=8549 mitochondria for control and 14725 for glutamate, from three independent experiments.

ROS. Rather, glutamate appears to act via converging EAAT1/AMPK-dependent and -independent pathways (Fig. 8).

## DISCUSSION

In this study, we discovered that glutamate reduces both lipid droplets and oxidative stress in astrocytes. There are several mechanisms to decrease the pool of lipid droplets. Our data reveal that lipolysis, mediated by the brain specific lipase DDHD2, does not seem to be enhanced but rather continues at a steady rate. We also found that lipophagy is unlikely to be responsible for the reduction in lipid droplets, as LC3B-associated autophagy is reduced. Similarly, there was no change in the colocalization of lipid droplets with lysosomes, which suggests that direct lysosome-based lipophagy, in which lipids are directly transferred from lipid droplets into lysosomes, is unlikely to be involved (Schulze et al., 2020). This is consistent with previous studies in which lysosomal inhibitors failed to prevent the reduction in astrocyte lipid droplets induced by glutamate agonists (Ioannou et al., 2019a).

Instead, our findings support a decrease in the formation of lipid droplets. Autophagy supplies lipids for lipid droplet growth (Rambold et al., 2015). Therefore, we speculate that the reduction in autophagy with glutamate treatment is responsible, at least in part, for the decrease in lipid droplets. This could explain the increase in mitochondrial mass in glutamate-treated astrocytes. Although this was not tested directly, we speculate that glutamate decreases mitophagy. This would serve to maintain a healthy pool of mitochondria capable of increasing catabolism of exogenous lipids released by neurons. An additional mechanism that remains to be tested is whether glutamate can also act directly on the lipid droplet biogenesis machinery. These two mechanisms could occur simultaneously to reduce the lipid droplet pool.

Consistent with findings in neurons (Marinangeli et al., 2018), this study reveals the activation of AMPK by glutamate. This is interesting because glutamate has many opposing functions in neurons versus astrocytes. For example, glutamate increases lipid droplets and oxidative stress in neurons and decreases mitochondrial mass (Ioannou et al., 2019a), whereas here we show that glutamate-induced AMPK activation reduces lipid droplets and confirm that AMPK activation decreases oxidative stress in astrocytes, as has been previously reported (Park et al., 2023; Xu et al., 2020; Yun et al., 2014). We also found that the effects on lipid droplets and ROS are independent of glutamate internalization. However, given that AMPK activation is blocked in the absence of glutamate internalization, this indicates that AMPK is sufficient, but not necessary, for glutamate-induced lipid droplet regulation. Rather, these two pathways appear to act synergistically.

Here, we also demonstrate that glutamate activates signaling pathways important for fatty acid oxidation downstream of AMPK. ACC2 converts acetyl-CoA into malonyl-CoA to inhibit fatty acid oxidation by preventing fatty acid entry into the mitochondria. Phosphorylation by AMPK inhibits ACC2 and promotes fatty acid oxidation. Therefore, the increased fatty acid oxidation, when exogenous lipids are present, is likely to be driven by glutamate-induced inactivation of ACC2. ACC1 similarly converts acetyl-CoA into malonyl-CoA, but this acts as the rate-limiting step in the synthesis of fatty acids. Phosphorylation of ACC1 by AMPK can decrease *de novo* fatty acid synthesis. As such, decreased fatty acid synthesis could also contribute to the reduction in lipid storage in droplets. Although our experiments do not distinguish between ACC1 and ACC2 isoforms, inhibition of either isoform is consistent with the reduction in lipid droplets observed, albeit via difference mechanisms. Whereas activation of AMPK in astrocytes has no effect on glutamate uptake, it decreases the entry of glutamate into the citric acid cycle (Voss et al., 2015). Perhaps, this helps astrocytes to favor fatty acid oxidation when it is available.

Of all the cell types in the brain, astrocytes possess the highest capacity for fatty acid metabolism (Fecher et al., 2019), but the function of catabolizing fatty acids extends beyond ATP production. It also serves to degrade neuronal fatty acids that might otherwise lead to lipotoxicity, and if those fatty acids contain oxidative modifications (i.e. lipid ROS), it could render neural cells susceptible to ferroptosis (Jacquemyn et al., 2024). This is important for preventing an inflammatory cascade affecting astrocytes and microglia (Mi et al., 2023). In addition to astrocytes, microglia also store peroxidated fatty acids derived from neurons in lipid droplets, and this exacerbates a proinflammatory phenotype (Li et al., 2024). Although this neuron-to-astrocyte transfer and subsequent degradation of fatty acids by astrocytes appears to be neuroprotective, the transfer of those same lipids to microglia disrupts their homeostatic functions and could contribute to disease (Bae et al., 2024). This might be due to the ability of astrocytes to coordinate the detoxification and degradation of lipid peroxides with the activity status of neurons. But when this process becomes disrupted, glia become lipid laden and contribute to disease.

Another interesting finding from this work is that glutamate regulates lipid droplet homeostasis independently of its internalization. This is important given that glutamate is a substrate for mitochondrial oxidation (McKenna, 2013), a finding that we confirmed here. This indicates that the effects on lipid droplets are not due to increased access to nutrients alone but to an unknown mechanism downstream of glutamate receptors. It will be interesting for future studies to elucidate which receptors, such as NMDA receptors (Ioannou et al., 2019b) or metabotropic glutamate receptors (Sun et al., 2013; Zeng et al., 2023), play a role in this process.

Finally, it is important to consider that the coupling of lipid metabolism between astrocytes and neural activity extends beyond pathology. Although our previous work shows that glutamate regulates lipid metabolism in serum-containing astrocyte cultures,

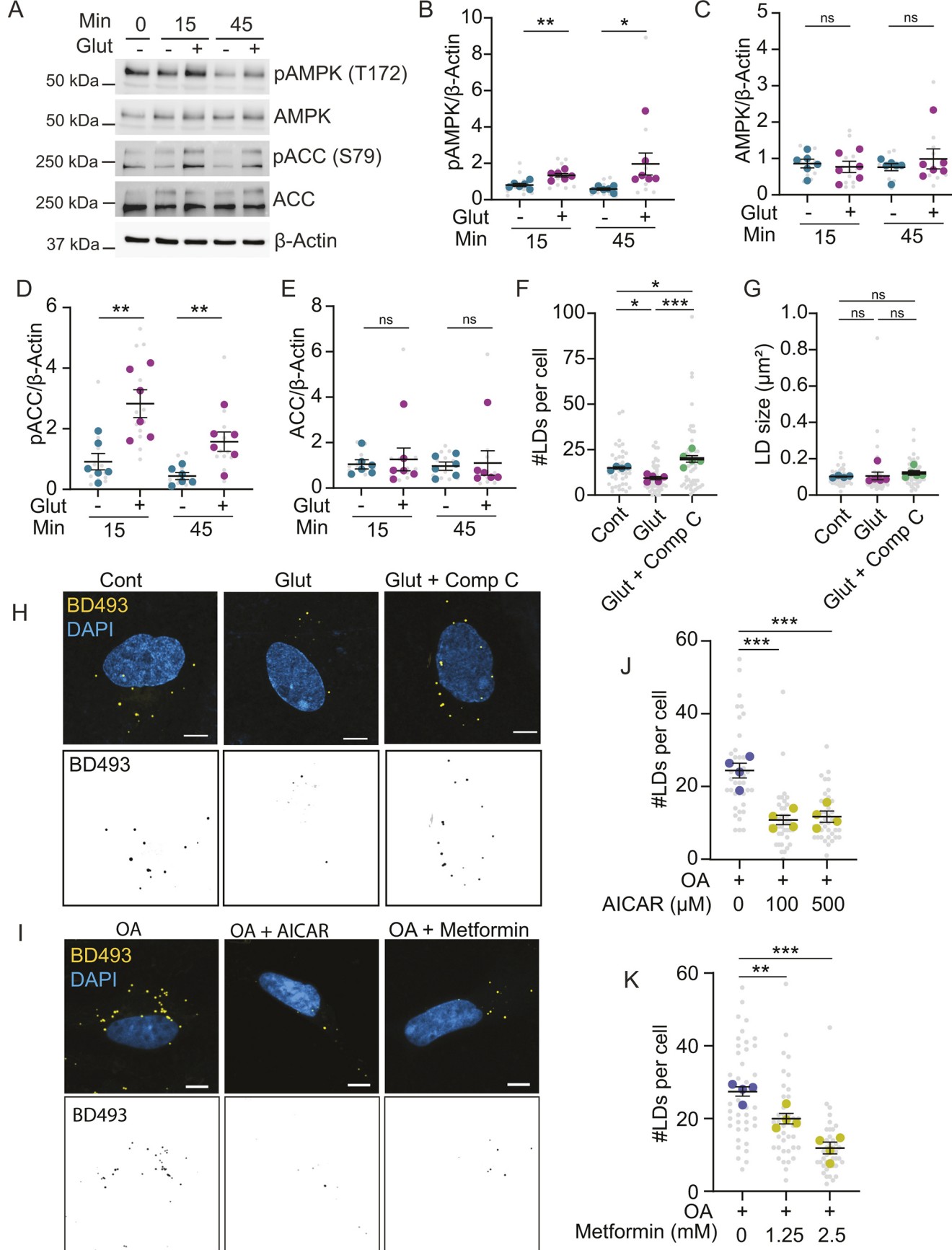

**Fig. 6.** See next page for legend.

**Fig. 6. Glutamate-induced AMPK signaling contributes to lipid droplet reduction.** (A) Astrocytes were treated with or without glutamate (glut) for the indicated times and analyzed by western blotting. (B–E) Quantification of phospho (p)-AMPK (B), AMPK (C), p-ACC (D) and ACC (E), normalized to β-actin. Mean±s.e.m. $n=6$ independent experiments, two technical replicates per n. Unpaired two-tailed $t$-test, *$P<0.05$, **$P<0.01$. (F–H) Astrocytes treated±glutamate and compound C (Comp C) in ABM. Cells were fixed, and lipid droplets were stained with BD493. Scale bars: 10 µm. Mean±s.e.m. $n=4$–5 independent experiments, ten technical replicates per $n$. One-way ANOVA with Tukey post-test, *$P<0.05$, ***$P<0.001$. (I) Astrocytes treated ±oleic acid and 100 µM AICAR or 2.5 mM metformin in ABM. Cells were fixed, and lipid droplets were stained with BD493. Scale bars: 10 µm. (J,K) Astrocytes treated±oleic acid and AICAR or metformin as indicated. Mean±s.e.m. $n=4$ independent experiments, nine to ten technical replicates per $n$. One-way ANOVA with Dunnett's post-test, **$P<0.01$, ***$P<0.001$. ns, non-significant.

the presence of serum induces a reactive phenotype typically associated with disease (Liddelow et al., 2017). This study using serum-free astrocytes suggests that these processes could also occur under more physiological states. Neurons transport lipids to glial lipid droplets during the day, which are cleared during sleep (Goodman et al., 2024; Haynes et al., 2024). This illustrates the importance of lipid transport from neurons to glia during physiological states. Perhaps glutamate is involved in stimulating detoxification and clearance of glial lipid droplets during sleep. Additional neurotransmitters and/or neuropeptides are also likely to be involved (Francés et al., 2024; Shoenhard and Sehgal, 2025).

A further consideration of this work is the concentration of glutamate used to elicit these effects. This study used 100 µM glutamate, as was previously shown to regulate astrocytic lipid droplets (Ioannou et al., 2019a). But the extracellular concentration of glutamate varies dramatically in the brain – typically, from 1 to 30 µM and climbing to 1 mM in the synapse following an action potential (Dzubay and Jahr, 1999; Kalivas, 2011). Whether these different pools of glutamate are sufficient to regulate lipid droplets and oxidative stress in astrocytes during health, or whether this metabolic coupling is more important during periods of excitotoxicity, has yet to be determined. What remains clear is that astrocytic fatty acid metabolism is intimately coupled to the activity of surrounding neurons with potential implications in both health and disease.

## MATERIALS AND METHODS
### Animal use
All experiments were approved by the Canadian Council of Animal Care at the University of Alberta (AUP#3358). Sprague-Dawley-timed pregnant rats were obtained from Charles River Laboratories and arrived at our facility 1 week before birth.

### Primary astrocyte culture
Primary hippocampal astrocyte cultures were prepared from postnatal day (P)0–P2 Sprague-Dawley rat pups as previously described (Ioannou et al., 2019b). Briefly, hippocampi were dissected and digested using papain (Worthington Biochemical Inc., LK003178) and benzonase nuclease (Sigma, E1014) for 25 min at 37°C, gently triturated and filtered with a 70-µm nylon cell strainer. Cells were grown in astrocyte-based medium (ABM) [neurobasal medium (Gibco, 21103) and Dulbecco's modified Eagle medium (DMEM) high glucose (Cytiva, SH30022FS), containing 2 mM Glutamax (Gibco, 35050061), 1 mM sodium pyruvate (Gibco, 11360070), 5 ng/ml heparin-binding epithelial growth factor (Sigma, E4643), 5 µg/ml N-acetyl-L-cysteine (Sigma, A8199), 100 µg/ml bovine serum albumin (BSA; Sigma, A2058), 100 µg/ml apo-transferrin human (Sigma, T1147), 16 µg/ml putrescine dihydrochloride (Sigma, P5780), 60 ng/ml progesterone (Sigma, P8783), 40 ng/ml sodium selenite (Sigma, S5261) and Antibiotic-Antimycotic (Gibco, 15240062)] or fetal bovine serum (FBS) medium [Basal Medium Eagle (Gibco, 21010046) containing

10% FBS (Gibco 10437028), 0.45% D-glucose (Sigma, G5146), 2 mM Glutamax, 1 mM sodium pyruvate and Antibiotic-Antimycotic]. Cells were plated on poly-D-lysine (Sigma, P6407)-coated coverslips for imaging experiments or culture dishes for all other experiments. Medium was changed after 20 min to reduce contaminating cell types in the cultures. Half medium was replaced every 2 days, and cells were used at days in vitro (DIV) 4–8.

### Immunofluorescence
Cells were fixed for 20 min with 4% paraformaldehyde (PFA) in PBS, blocked and permeabilized for 1 h in 2% BSA+0.2% Triton X-100 in PBS, then incubated for 1 h in primary antibodies at room temperature [chicken polyclonal anti-GFAP (Abcam, ab4674, RRID: AB_304558; 1:500), rabbit monoclonal anti-TOMM20 (Abcam, ab186735, RRID: AB_2889972; 1:500)] or fixed in ice-cold methanol [rabbit polyclonal anti-LC3B (Abcam, ab48394, RRID: AB_881433; 1:750)]. Cells were washed with 0.2% Triton X-100 in PBS and then incubated for 1 h in the secondary antibodies: goat anti-chicken AlexaFluor647 (Thermo Fisher Scientific, A21449, RRID: AB_2535866; 1:750), donkey anti-rabbit AlexaFluor647 (Thermo Fisher Scientific, A31573, RRID: AB_2536183; 1:750) or goat anti-rabbit AlexaFluor488 (Thermo Fisher Scientific, A11008, RRID: AB_143165; 1:750). Cells were washed, stained with DAPI (Abcam, ab228549), washed and mounted in DAKO Medium (Agilent, S302380-2).

### Microscopy
Imaging was performed using a LSM900 Airyscan2 (Carl Zeiss) equipped with a plan-apochromat 63× oil objective (1.4 NA) with ZEN software or a Leica Stellaris 5 with a plan-apochromat 63× oil objective (1.4 NA) with LAS X software.

### Western blotting
Astrocytes were treated with or without 100 µM glutamate (Cayman, 30377) and/or 200 nM (3S)-3-[[3-[[4-(trifluoromethyl) benzoyl] amino] phenyl] methoxy]-L-aspartic acid (TFB-TBOA; Tocris, 2532) for the indicated time at 37°C, washed three times in PBS and lysed in 20 mM HEPES pH 7.4, 100 mM NaCl, 1% Triton X-100, 5 mM EDTA, Halt Protease and Phosphatase Inhibitor (Thermo Fisher Scientific, 78440), resolved by SDS-PAGE, and processed for western blotting. Antibodies used were as follows: rabbit monoclonal anti-AMPKα (D63G4) (Cell Signaling Technology, 5832, RRID: AB_10624867; 1:1000), rabbit monoclonal anti-phospho-AMPKα Thr172 (40H9) (Cell Signaling Technology, 2535, RRID: AB_331250; 1:1000), rabbit monoclonal anti-acetyl-CoA carboxylase 1/2 (C83B10) (Cell Signaling Technology, 3676, RRID: AB_2219397; 1:1000), rabbit monoclonal anti-phospho-acetyl-CoA carboxylase 1/2 Ser79 (Cell Signaling Technology, 3661, RRID: AB_330337; 1:1000), mouse monoclonal anti-β-actin (Cell Signaling Technology, 3700, RRID: AB_2242334; 1:10,000), rabbit polyclonal anti-LC3B (Abcam, ab48394, RRID: AB_881433; 1:1000), goat polyclonal anti-mouse HRP (Invitrogen, A16072, RRID: AB_2534745; 1:10,000) and goat polyclonal anti-rabbit HRP (Invitrogen, A16104, RRID: AB_2534776; 1:10,000). Densitometry was performed using ImageJ. Blots were cropped for main figures. Corresponding full blots can be found in Fig. S1.

### Intracellular glutamate uptake
Astrocytes were plated on 10-cm plates and treated with 100 µM glutamate and increasing concentrations of TFB-TBOA as indicated for 15 min at 37°C in ABM. Cells were washed twice in PBS, lysed and analyzed in duplicate using a Glutamate Assay Kit (Abcam, ab83389) and a Synergy Mx Multi-Mode Microplate Reader (BioTek Instruments Inc). Glutamate levels were normalized to total protein levels as determined by bicinchoninic acid (BCA) assay (Pierce, 23225) run in parallel.

### Lipid droplet analysis
Astrocytes were treated with or without 100 µM glutamate and 200 nM TFB-TBOA, 2 µM KLH45 (Sigma, SML1998), 5 µM dorsomorphin dihydrochloride (compound C; Abmole, M2238), or 2 µM PF-04620110 (Sigma, PZ0207) and 10 µM PF-06424439 (Cayman, 17680) in ABM for 4 h at 37°C. Astrocytes were treated with 24 µM oleic acid-BSA (Sigma,

Journal of Cell Science

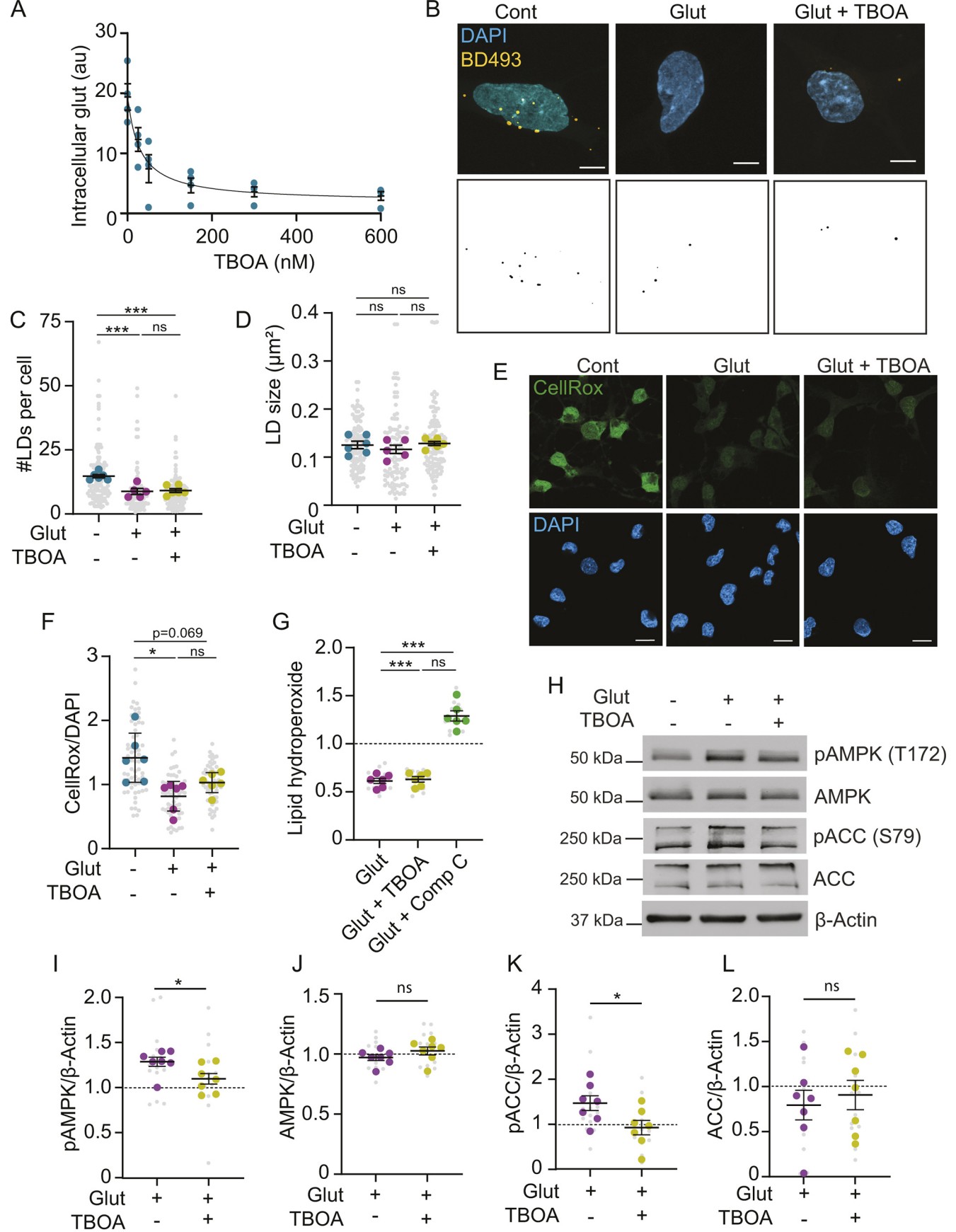

**Fig. 7.** See next page for legend.

**Fig. 7. Glutamate reduces lipid droplets and reactive oxygen species independently of internalization.** (A) Relative intracellular glutamate (glut) levels of astrocytes treated with glutamate and increasing concentrations of the EEAT1/2 inhibitor TFB-TBOA (TBOA). au, arbitrary units. Mean±s.e.m. $n=4$ independent experiments. (B–D) Astrocytes treated±glutamate and TFB-TBOA in ABM. Cells were fixed, and lipid droplets were stained with BD493. Scale bars: 10 µm, Mean±s.e.m. $n=5$–6 independent experiments, 20 technical replicates per $n$. One-way ANOVA with Tukey post-test, ***$P<0.001$. (E,F) Astrocytes treated±glutamate and 200 nM TFB-TBOA in ABM and labeled with CellRox Green were imaged (E) and quantified (F). Scale bars: 10 µm. Mean±s.e.m. $n=6$ independent experiments, ten technical replicates per $n$. One-way ANOVA with Tukey's post-test, *$P<0.05$. (G) Astrocytes were treated with or without glutamate, TFB-TBOA or compound C and analyzed for lipid hydroperoxides. Values normalized to no treatment control (dashed line) Mean±s.e.m. $n=6$ independent experiments, three technical replicates per $n$. One-way ANOVA with Tukey post-test, ***$P<0.001$. (H) Astrocytes were treated with or without glutamate and TFB-TBOA for 15 min and analyzed by western blotting. (I–L) Quantification of p-AMPK (I), AMPK (J), p-ACC (K) and ACC (L), normalized to β-actin. Mean ±s.e.m. $n=7$ independent experiments, two technical replicates per $n$. Unpaired two-tailed $t$-test, *$P<0.05$. ns, non-significant.

O3008) overnight followed by 100 µM glutamate in ABM for 3 h at 37°C or 24 µM oleic acid-BSA together with 100 µM or 500 µM AICAR (Sigma, A9978), or 1.25 mM or 2.5 mM metformin (Sigma, 317240) in ABM for 4 h at 37°C. Cells were washed in Dulbecco's phosphate buffered saline (DPBS; Fisher Scientific, SH3002803), fixed in 4% PFA and stained with 5 µg/ml BODIPY 493/503 (BD493; Thermo Fisher Scientific, D3922) for 1 h at room temperature followed by DAPI staining and mounting as described above. Z-stacks (0.4 µm) were obtained and analyzed in ImageJ. Maximum-intensity projections were background subtracted and thresholded using Renyi Entropy, followed by water shedding. Particle average size and number (5-infinity pixels) were analyzed. For DGAT inhibitor and oleic acid experiments, images were thresholded using Renyi Entropy, and particle average size and number (7–240 and 7–320 pixels, respectively) with 0.9–1.0 circularity were analyzed.

### Lysotracker colocalization analysis

Astrocytes were treated with 100 µM glutamate and 24 µM oleic acid-BSA in ABM for 4 h at 37°C. After 3 h and 30 min, LysoTracker™ Red DND-99 (Invitrogen, L7528) was added to the ABM. The cells were incubated for additional 30 min, washed in DPBS, fixed in 4% PFA and stained with 5 µg/ml BD493 (Thermo Fisher Scientific, D3922) for 1 h at room temperature followed by DAPI staining and mounting as described above.

Images were analyzed in ImageJ, where each channel was thresholded using Renyi Entropy, a mask of colocalization was generated using the 'AND' function, and percentage colocalization was indicated as the area of overlap relative to the total area of each channel. For Lysotracker, particle number (5-infinity pixels) was also analyzed.

### LC3B analysis

Astrocytes were treated with or without 100 µM glutamate and/or 100 µM bafilomycin A1 (Sigma, C974N95), for 4 h at 37°C in ABM Cells were washed, lysed and processed for western blotting as described above. Densitometry was performed using ImageJ. Autophagy induction was calculated as the LC3B-II/LC3B-I ratio in the presence of bafilomycin A1, and autophagic flux was determined as the LC3B-II/LC3B-I ratio in the presence of bafilomycin A1 minus the LC3B-II/LC3B-I ratio in the absence of bafilomycin A1. Cells were fixed in ice-cold methanol for 10 min, immunostained for LC3B, mounted and imaged as described above. Z-stacks (0.5 µm) were obtained and analyzed in ImageJ. Maximum-intensity projections were background subtracted and thresholded using Renyi Entropy. Particle average size and number (5-infinity pixels) were analyzed.

### Mitochondrial lipid trafficking assay

Astrocytes were incubated in 2 µM BODIPY 558/568 C12 (Red-C12) (Thermo Fisher Scientific, D3835) in ABM overnight, washed three times in DPBS (Fisher Scientific, SH3002803) to remove remaining Red-C12 and treated with or without 100 µM glutamate (Cayman, 30377) in ABM for 4 h at 37°C. 500 nM MitoTracker Deep Red (Thermo Fisher Scientific, M22426) was added for the final 30 min. Cells were washed three times in DPBS, fixed in 4% PFA, stained with DAPI (Abcam, ab228549) and imaged as described above. Images were analyzed using ImageJ. Images were background subtracted (rolling ball radius 25), and a binary mask of MitoTracker Deep Red Channel was used to create an ROI in the Red-C12 channel. The mean intensity of Red-C12 in the mitochondria was analyzed.

### CellRox assay

Astrocytes were treated with or without 100 µM glutamate and 200 nM TFB-TBOA in ABM for 4 h at 37°C. 2.5 µM CellRox Green (Invitrogen, C10444) was added for the final 30 min. Cells were fixed, stained for DAPI, mounted and imaged as described above. Images were analyzed in ImageJ. A mask of the DAPI channel was generated using Otsu thresholding, and the mean intensity of CellRox Green channel was normalized to the mean intensity of DAPI.

### Lipid peroxidation assay

Astrocytes in 60-mm plates were treated with or without 100 µM glutamate and 200 nM TFB-TBOA or 5 µM compound C in ABM for 6 h at 37°C.

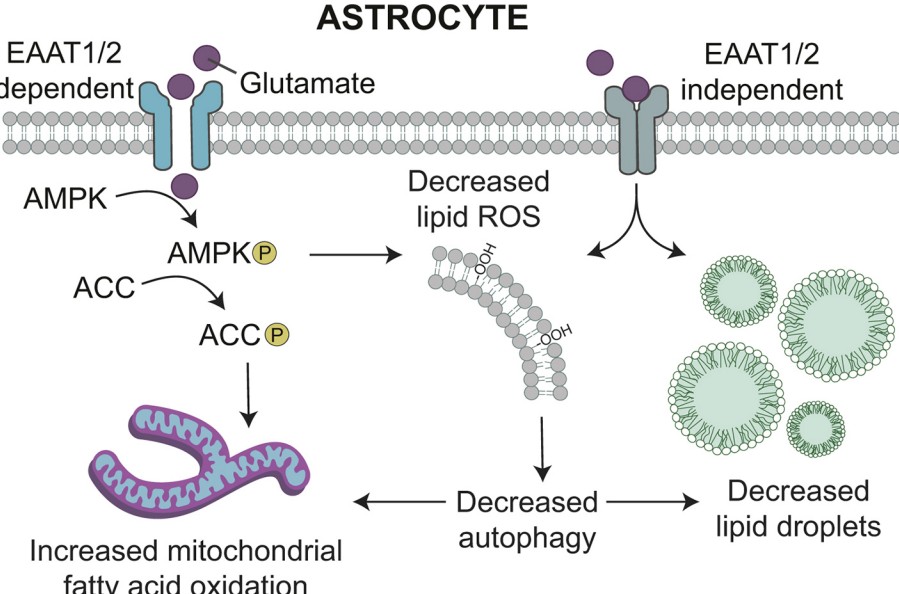

**ASTROCYTE**

**Fig. 8. Model of lipid regulation by glutamate.** Glutamate reduces lipid droplet biogenesis and enhances mitochondrial fatty acid catabolism. This is likely due to its ability to reduce reactive oxygen species (ROS), which would (1) maintain the health of mitochondria to support increased fatty acid oxidation and (2) decrease the need to store lipids generated by autophagy in lipid droplets. Glutamate exerts these effects via converging EAAT1/2/AMPK-dependent and -independent pathways.

Cells were washed in ice-cold PBS, scraped into ice-cold high-performance liquid chromatography-grade water and sonicated for 30 s. Lipid hydroperoxides were extracted from cell lysates and analyzed using a lipid hydroperoxide assay kit (Abcam, ab133085), according to the manufacturer's protocol, and Synergy Mx Multi-Mode Microplate reader (BioTek Instruments Inc.).

## Analysis of mitochondrial morphology

Astrocytes were treated with or without 100 µM glutamate for 4 h in ABM. 30 min before completing the experiment, astrocytes were incubated in 5 µM CellTracker Green CMFDA Dye (Thermo Fisher Scientific, C2925). Cells were then fixed with 4% PFA, blocked and immunostained for TOMM20 as described above. Z-stacks (0.4 µm) were obtained, imaged as described above and analyzed in ImageJ. The CellTracker Green channel was thresholded using Li and used to create an ROI. The 3D Object Counter function was used to determine the cell volume. TOMM20 within the ROI was threshold using IsoData, and the 3D Object Counter Function was used to determine the average number, length and volume of mitochondria.

## Metabolic flux analyses

Astrocytes were plated on Xfe96/XF Pro PDL-coated cell culture microplates. Astrocytes were incubated with 5 µM CellTracker Green for 15 min in ABM and washed twice in substrate-limited medium [DMEM medium supplemented with 2 mM glucose, 0.5 mM L-carnitine (Thermo Fisher Scientific, 24104001) and 5 ng/ml heparin-binding epithelial growth factor]. The Seahorse XF Long Chain Fatty Acid Oxidation Stress Test (Agilent, 103672-100), Seahorse XF Real-Time ATP Rate Assay (Agilent, 103592-100) or Seahorse XF Mito Fuel Flex Assay (Agilent, 103270-100) were performed according to the manufacturer's protocol using an Agilent Seahorse xFe96 Metabolic Analyzer as previously described (Mi et al., 2023; Qi et al., 2021; Weightman Potter et al., 2019). In brief, astrocytes were incubated in substrate-limited medium for 1 h at 37°C in a $CO_2$-free incubator. For Mito Fuel Flex Assays, respiration was assessed in the presence or absence of fuel pathway inhibitors: 2 µM UK5099 for pyruvate; 3 µM BPTES and 200 nM TFB-TBOA for glutamate/glutamine; 20 µM etomoxir for fatty acids.

Calculations used were as follows (OCR, oxygen consumption rate): % dependency=[(baseline OCR−target inhibitor of OCR)/(baseline OCR−all inhibitors of OCR)]×100%; % capacity=1/[(baseline OCR−other two inhibitors of OCR)/(baseline OCR−all inhibitors of OCR)]×100%. Fuel flex calculations were performed on the means of the technical replicates. For fatty acid oxidation stress assays, substrate-limited medium was supplemented with or without 200 µM BSA-palmitate (Cayman, 29558) and/or 20 µM etomoxir. After five baseline measurements, astrocytes were treated with 100 µM glutamate for 4 h, and metabolic measurements were assessed every 3 min. After glutamate treatment, astrocytes were sequentially treated with 1.5 µM oligomycin, 1 µM FCCP and, finally, 0.5 µM rotenone/antimycin A (all as supplied in the Seahorse kit, Agilent). Metabolic measurements were normalized to cell number as indicated by CellTracker Green fluorescence intensity measured using a Synergy Mx Multi-Mode Microplate Reader (BioTek Instruments Inc). There were 10–12 replicates per treatment condition, and each average measurement was normalized to the last baseline measurement.

## Metabolomic profiling

Astrocytes in 15-cm plates were treated with or without 100 µM glutamate for 1 h 45 min in ABM. Cells were washed in PBS, scraped into 1 ml ice-cold methanol, flash frozen and processed by The Metabolomics Innovation Center, Canada. Samples were vortexed, sonicated and centrifuged at 21,000 $g$ for 10 min. The protein pellets were used to determine protein concentration using standard BCA assay. The supernatants were analyzed for medium- to long-chain fatty acids (Han et al., 2015). Standards of the targeted fatty acids were prepared in liquid chromatography-mass spectrometry-grade methanol from 0.00001 to 0.5 µg/ml. 200 µl of the supernatant or standard was dried in a speed vacuum concentrator. 30 µl 100 mM 3-nitrophenylhydrazine solution and 30 µl 100 mM EDC-3% pyridine solution were added. The mixtures were incubated at 50°C for 60 min. After reaction, 6-µl aliquots of the resultant solutions were injected to run ultra-performance liquid chromatography multiple reaction

monitoring mass spectrometry (UPLC-MRM/MS) with a C8 column (2.1×50 mm, 1.8 µm) for liquid chromatography separation, with the use of 0.01% formic acid in water and 1:1 acetonitrile-isopropanol as the mobile phase for gradient elution (20% to 100% in 18 min), at 0.35 ml/min and 50°C. Quantitation of carnitines was carried out as in Han et al. (2018). Standard solutions of targeted carnitines were prepared in 0.00001 to 1 µM methanol. 200 µl of supernatant or standard was dried and resuspended in 25 µl 100 mM 3-nitrophenylhydrazine solution and 25 µl 100 mM EDC-3% pyridine solution. The mixtures were incubated at 40°C for 30 min. 6 µl solution was injected to run UPLC-MRM/MS with positive-ion detection. An Agilent 1290 UHPLC system coupled to an Agilent 6595B QQQ instrument operated in the negative-ion multiple-reaction monitoring (MRM)/MS mode was used for both analyses. Log2 fold change of each fatty acid and acylcarnitine species was calculated as mean (pmol/mg of protein) of the glutamate-treated ABM-cultured astrocytes/mean (pmol/mg protein) of the control ABM-cultured astrocytes, and the resulting values were Log2 transformed. Raw data are provided in Table S1.

## Statistical analysis

Statistical analysis was performed using GraphPad Prism 10. All graphs are depicted as SuperPlots in which independent replicates are shown in large shapes, and the technical replicates are shown as small gray shapes (Lord et al., 2020). The independent replicate values were calculated from the mean of technical replicates. Statistical analyses were performed on the independent replicates for each experiment. Statistical tests used are indicated in the corresponding figure legend.

### Acknowledgements
Experiments were performed at the University of Alberta Faculty of Medicine & Dentistry Cell Imaging Core, RRID:SCR_019200. We thank Donald Van Meyel and Jishang Jiang for helpful comments on the manuscript; Fei Yin, Guoyuan Qi and Mike Wong for advice on performing metabolic flux assays; Jun Han from The Metabolomics Innovation Center for performing the analysis of fatty acids and acylcarnitines; Vivek Kumar for images from SciDraw; and Marnie-Maddock for images from Bioicons.

### Competing interests
The authors declare no competing or financial interests.

### Author contributions
Conceptualization: L.F.R.-A., J.J., M.S.I.; Data curation: L.F.R.-A., J.C., J.J., I.R., I.I., M.S.I.; Formal analysis: L.F.R.-A., J.C., J.J., I.R., I.I., M.S.I.; Funding acquisition: M.S.I.; Investigation: L.F.R.-A., J.C., J.J., I.R., I.I., M.S.I.; Methodology: M.S.I.; Project administration: M.S.I.; Supervision: M.S.I.; Writing – original draft: M.S.I.; Writing – review & editing: L.F.R.-A., J.C., J.J., I.R., I.I., M.S.I.

### Funding
This work was supported by a Natural Sciences and Engineering Research Council of Canada Discovery Grant (#2020-04047) and the Canada Research Chairs Program (#2021-00027). L.F.R.-A. was supported by a Graduate Studentship Award from the Women and Children's Health Research Institute. I.R. was supported by a Canada Graduate Scholarship from the Canadian Institutes of Health Research Doctoral Award (#181551) and an Izaak Walton Killam Memorial Scholarship from the University of Alberta. J.J. was supported by an EMBO postdoctoral fellowship (ALTF 120-2022). Open Access funding provided by University of Alberta. Deposited in PMC for immediate release.

### Data and resource availability
All relevant data and details of resources can be found within the article and its supplementary information.

### Peer review history
The peer review history is available online at https://journals.biologists.com/jcs/lookup/doi/10.1242/jcs.263983.reviewer-comments.pdf

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
