## [Peer Review File · Journal of Cell Science]

Glutamate decreases oxidative stress and lipid droplet formation in astrocytes

Luis F. Rubio-Atonal, Jinlan Chang, Julie Jacquemyn, Isha Ralhan, Iset Ilarraza and Maria Ioannou

DOI: 10.1242/jcs.263983

Editor: Subhojit Roy

Review timeline

Original submission:	5 March 2025
Editorial decision:	2 April 2025
First revision received:	5 August 2025
Accepted:	2 September 2025

Original submission

First decision letter

MS ID#: jcs.263983

MS TITLE: Glutamate signaling decreases astrocytic lipid droplets and oxidative stress

AUTHORS: Luis F Rubio-Atonal; Jinlan Chang; Isha Ralhan; Julie Jacquemyn; Maria Ioannou

ARTICLE TYPE: Research Article

Dear Dr Ioannou,

We have now reached a decision on the above manuscript.

To see the reviewers' reports and a copy of this decision letter, please go to:

As you will see, the reviewers raise a number of substantial criticisms that prevent me from accepting the paper at this stage. They suggest, however, that a revised version might prove acceptable, if you can address their concerns which may be addressed by a few largely feasible experiments and text changes.

If you think that you can deal satisfactorily with the criticisms on revision, I would be pleased to see a revised manuscript.

Reviewer 1

SUMMARY OF THE ADVANCE MADE IN THIS PAPER AND ITS POTENTIAL SIGNIFICANCE TO THE FIELD

This is a cell biological study investigating the mechanisms by which glutamate causes a reduction in lipid droplets in astrocytes. Astrocytes serve as important recipient depots for excess neuronal lipids to combat neurodegenerative disease. In a previous study (Ioannou et al, 2019), it was observed that glutamate can directly act on astrocytes to reduce LD numbers, while concomitantly increasing mitochondrial \dot{I}^2 -oxidation, but the mechanism was not defined. Here, the authors demonstrate that glutamate increases \dot{I}^2 -oxidation in primary astrocytes even in serum-free medium, suggesting an endogenous mechanism. First, the authors demonstrated basal lipolysis in astrocytes via the lipase DDHD2, but glutamate stimulation of lipolysis was ruled out. Lipophagy was also not

implicated in glutamate-induced LD loss, as there was reduced autophagic flux after glutamate treatment. Instead, glutamate caused a reduction in astrocyte ROS/lipid peroxidation via AMPK activation, independent of glutamate internalization. This glutamate-AMPK axis was also necessary for LD loss in astrocytes, likely by reducing LD biogenesis from autophagy-derived FA's.

SUGGESTIONS TO AUTHORS

*Stimulation of lipolysis by glutamate was ruled out due to lack of change in LD size and no increase in FA's or acylcarnitines. This may be true, but given the low abundance of LDs in the first place, these parameters may not change noticeably due to lipolysis alone (especially the reduction in LD size, when there are ~15 LDs/cell). Therefore, it may be more illuminating to measure the rate of LD loss over a time course, $\Delta \pm$ glutamate in the presence of an LD synthesis inhibitor such as Triacsin C or DGAT1/2 inhibitors.

*The autophagic flux experiments convincingly show a decrease in glutamate-treated cells. However, autophagic flux does not always equate to lipophagy, per se. Do LDs colocalize with lysosome markers (lysotracker, Lamp1, or CD63 for example) after glutamate treatment? Is this exacerbated further by BafA1 treatment?

*The findings that glutamate elicits effects independent of intracellular transport are very interesting. Does TBOA have any effect on autophagic flux, AMPK activation, or phosphorylation of ACC?

*The authors show that AMPK activity is necessary for LD loss in astrocytes. Does AMPK activation alone cause LD loss, or is glutamate still needed for this? In other words, if AMPK is necessary, is it also sufficient?

*The authors posit that glutamate reduction in autophagic flux reduces LD biogenesis (with a possible reduction in de novo lipogenesis as well). But glutamate may also directly affect the LD biogenesis machinery. Do glutamate-treated cells grow LDs more slowly upon lipid loading? If not, this could further support the original claim of FA supply, and not TAG production.

*Minor - Upon a causal first read, the first two sentences of the abstract appeared to be conflicting. It eventually became clear, but it might be worth revising these for clarity.

Reviewer 2

SUMMARY OF THE ADVANCE MADE IN THIS PAPER AND ITS POTENTIAL SIGNIFICANCE TO THE FIELD

The manuscript by Rubio-Atonal et al. explores the effect of glutamate on fatty acid metabolism in astrocytes. The work is a follow-up on the very interesting and influential study by Ioannou et al. (2019), which showed that astrocytes take up FAs released from neurons and store them in lipid droplets (LDs) or consume via mitochondrial β^2 -oxidation, protecting neurons from FA toxicity and oxidative stress during neuronal activity. As glutamate is the main excitatory neurotransmitter in the brain, studying its effect on astrocytes represents a simplified system to understand how astrocytes respond to neuronal activity. The authors show that glutamate activates the AMPK signaling pathway in primary hippocampal rat astrocytes, and that it is not taken up by the cells. They observe a decrease in astrocyte LD size and a reduction in autophagy, suggesting that these processes are regulated by AMPK signaling, although they do not explain how. They propose that the reduced autophagy leads to a decrease in LDs, although again the causality and the temporal order of events are not established.

SUGGESTIONS TO AUTHORS

This manuscript addresses a very interesting question, pertinent to understanding the neuron/astrocyte interaction during sleep-wake cycle or in pathological states using a primary astrocyte cell model. This is surely a demanding experimental system that is not easy to manipulate. A number of processes are evaluated, often with a single experiment and a number of assumptions are not well justified, so the final model is not very convincing. A more critical

assessment by the authors is needed. No such assessment is provided in the Discussion; the Discussion should be thoroughly revised.

Specific comments:

1. The model used in the study are primary rat astrocytes cultured in serum-free medium, which is to maintain them in quiescent state. Why is it important to maintain quiescent state, and is this state relevant for the understanding of metabolic coupling between neurons and astrocytes? Furthermore, the authors use glutamate at 100 μM ; is this concentration physiologically relevant? More explanation regarding the chosen experimental conditions would be useful.
2. In Fig 2, mitochondrial activity is assessed using Seahorse assays, and in combination with different inhibitors to determine which fuel is used by the mitochondria (pyruvate, glutamate/glutamine or fatty acids). No reference is given for these inhibitors, and no control is provided to assess their activity or specificity, so it's not possible to judge how reliable are these results. Only a single inhibitor per pathway is used. In general, the study relies heavily on the use of inhibitors and we have to trust that the inhibitors function as stated. TFB-TBOA is the most problematic because it has no effect and no information is given, compound C is inhibitory, but is it specific? At the least, more information is needed, whereas alternative approaches and controls would strengthen the conclusions.
3. In all figures, the authors state the number of independent experiments and the number of replicates per experiment, but it is not stated how many cells were analyzed in each replicates, and therefore what was the n used for statistical analyses. This needs to be clarified in all figure legends and the statistical tests used should be better justified.
4. The authors show that glutamate decreases LD number, but not LD size. However, the effect on LD number varies between figures: 3-fold decrease in Fig.3 ($p < 0.001$) vs 2-fold in Fig. 4 ($p < 0.05$) vs. 1.5-fold in Fig. 7E ($p < 0.05$). As they are performed now, the statistical tests don't seem to be very reliable.
 - The authors argue that this decrease is not due to an increase in lipolysis because LD size is not affected, citing a study in liver cells; what is the evidence that liver cells are a relevant comparison?
 - The other argument is that free fatty acids are not increased, but there actually is some increase in free fatty acids (fig. 4D), except it is not assessed whether this increase is statistically significant. There is a large variability between experiments; plotting data in super plots would allow visual comparison between individual experiments. Taking into account the variability between experiments, might there be a difference in each experiment between +/- glutamate conditions? And how much of an increase in free FAs can be expected in these cells if lipolysis is induced?
5. Mitochondrial volume is increased (Fig. 5); what is the connection with lipid droplets and oxidative metabolism? Contradicting statements seem to be given in different parts of the manuscript.

First revision

Author response to reviewers' comments

Comments from the Reviewers:

Reviewer 1: SUMMARY OF THE ADVANCE MADE IN THIS PAPER AND ITS POTENTIAL SIGNIFICANCE TO THE FIELD

This is a cell biological study investigating the mechanisms by which glutamate causes a reduction in lipid droplets in astrocytes. Astrocytes serve as important recipient depots for excess neuronal lipids to combat neurodegenerative disease. In a previous study (Ioannou et al, 2019), it was observed that glutamate can directly act on astrocytes to reduce LD numbers, while concomitantly increasing mitochondrial β -oxidation, but the mechanism was not defined. Here, the authors demonstrate that glutamate increases β -oxidation in primary astrocytes even in serum-free medium, suggesting an endogenous mechanism. First, the authors demonstrated basal lipolysis in astrocytes via the lipase DDHD2, but glutamate stimulation of lipolysis was ruled out. Lipophagy was also not implicated in glutamate-induced

LD loss, as there was reduced autophagic flux after glutamate treatment. Instead, glutamate caused a reduction in astrocyte ROS/lipid peroxidation via AMPK activation, independent of glutamate internalization. This glutamate-AMPK axis was also necessary for LD loss in astrocytes, likely by reducing LD biogenesis from autophagy-derived FA's.

SUGGESTIONS TO AUTHORS

*Stimulation of lipolysis by glutamate was ruled out due to lack of change in LD size and no increase in FA's or acylcarnitines. This may be true, but given the low abundance of LDs in the first place, these parameters may not change noticeably due to lipolysis alone (especially the reduction in LD size, when there are ~15 LDs/cell). Therefore, it may be more illuminating to measure the rate of LD loss over a time course, \pm glutamate in the presence of an LD synthesis inhibitor such as Triacsin C or DGAT1/2 inhibitors.

This is an excellent suggestion. We repeated the experiment in the presence of DGAT1/2 inhibitors as suggested and found that glutamate no longer affects LD numbers. This new data can be found in Fig 4G-I of the revised manuscript and referred to on Lines 133-137 “we tested whether glutamate continues to reduce lipid droplets in the absence of triglyceride synthesis by inhibiting diacylglycerol acyltransferase (DGAT) 1 and 2. We reasoned that if lipolysis was increased, then glutamate would further reduce lipid droplets in the absence of DGAT1/2 activity. However, glutamate had no additional effect on lipid droplets in the presence of DGAT-1 and 2 inhibitors (Fig. 4G-I).” We were concerned that a time course would not be sensitive enough to interpret given the low number of lipid droplets, as the reviewer points out.

Instead, we expanded our lipidomics analysis to show individual fatty acid species following glutamate treatment. We reasoned that the high sensitivity of UPLC-MRM/MS would detect subtle changes in individual species that could have been masked when we had grouped all fatty acids species together. However, no changes in any free fatty acid species upon glutamate treatment were observed. This new data can be found in Fig 4F of the revised manuscript. We thank the reviewer for the suggestion as these new data further support that glutamate is acting on LD formation as opposed to lipolysis.

*The autophagic flux experiments convincingly show a decrease in glutamate-treated cells. However, autophagic flux does not always equate to lipophagy, per se. Do LDs colocalize with lysosome markers (lysotracker, Lamp1, or CD63 for example) after glutamate treatment? Is this exacerbated further by BafA1 treatment?

This is also a great suggestion. We performed new experiments to assess co-localization of LDs with lysosomes stained with lysotracker. We found that LDs do not increase co-localization with lysosomes upon glutamate treatment as would be expected if lipophagy was induced. This new data can be found in **Fig. 5G-J** of the revised manuscript and on Lines 153-160 we state “Lipophagy can also occur through the docking of lysosomes onto lipid droplets and the direct transfer of neutral lipids into the lysosomes (Schulze et al., 2020). To test this, we assessed co-localization between BD493-positive lipid droplets and acidic compartments stained with LysoTracker. We found no difference in the percent co-localization relative to BD493 with glutamate while the percent co-localization relative to LysoTracker decreased (Fig. 5G-I). Glutamate did not affect the number of LysoTracker-positive puncta (Fig. 5J). This indicates that glutamate does not increase the association of lipid droplets with lysosomes as would be required for lipophagy.”

We were unable to assess this co-localization with BafA1 as this neutralizes lysosomal pH which prevents lysotracker from fluorescing. Instead, we expanded our discussion reference our previous work showing that BafA1 treatment does not affect glutamate-induced reduction in LDs, as would be expected in lipophagy was responsible. On Lines 225-230, we now state “Similarly, there was no change in the co-localization of lipid droplets with lysosomes which suggests that direct lysosome-based lipophagy, where lipids are directly transferred from lipid droplets into lysosomes, is unlikely to be involved (Schulze et al., 2020). This is consistent with previous studies where lysosomal inhibitors fail to prevent the reduction in astrocyte lipid droplets induced by glutamate agonists (Ioannou, Jackson, et al., 2019a).”

We believe these results support the conclusions that lipophagy is unlikely to be responsible for the reduction in lipid droplets downstream of glutamate and we thank the reviewer for the suggestion.

*The findings that glutamate elicits effects independent of intracellular transport are very interesting. Does TBOA have any effect on autophagic flux, AMPK activation, or phosphorylation of ACC?

We performed new experiments as recommended by the reviewer. It seems that AMPK and ACC activation are dependent on glutamate transport as TFB-TBOA prevents the increase in phosphorylation. This new data can be found in Fig 7H-L of the revised manuscript and Lines 209- 214 “we tested whether the activation of AMPK is also independent of glutamate uptake. However, TFB-TBOA prevented the activation of both AMPK and ACC by glutamate, with no changes in total levels of protein (Fig. 7H-L). Collectively, this data indicates that AMPK activation is sufficient, but not necessary for the effects of glutamate on lipid droplets and ROS. Rather, glutamate appears to act via converging EAAT1/AMPK dependent and independent pathways (Fig. 8).”

These results were unexpected given that the reduction in ROS and LDs are transport independent and yet AMPK activation is sufficient to reduce LDs (see comment below) and ROS. In light of this new information, we have also revised our model in Fig 9 to indicate that glutamate acts on LDs via EAAT-dependent and independent mechanisms. We are grateful to the reviewer for pushing us to explore this further as these new experiments reveal a more nuanced mechanism than we had previously appreciated.

*The authors show that AMPK activity is necessary for LD loss in astrocytes. Does AMPK activation alone cause LD loss, or is glutamate still needed for this? In other words, if AMPK is necessary, is it also sufficient?

We performed new experiments as recommended by the reviewer. We treated astrocytes with oleic acid and two different AMPK activators, AICAR or metformin and observed a reduction in the number of LDs with both drugs. This new data can be found in **Fig 6I-K** of the revised manuscript and lines 187-190 “*Conversely, AMPK activators, AICAR and Metformin, reduce the number of lipid droplets in the absence of glutamate (Fig. 6I-K). These data indicate that glutamate activates AMPK in astrocytes which can contribute to the reduction in lipid droplets upon glutamate treatment.*” This new data combined with the experiments performed above reveal that AMPK activation is sufficient but not necessary for LD loss and likely indicates a converging mechanism. As stated above, this is reflected in our revised model in **Fig 9**.

*The authors posit that glutamate reduction in autophagic flux reduces LD biogenesis (with a possible reduction in de novo lipogenesis as well). But glutamate may also directly affect the LD biogenesis machinery. Do glutamate-treated cells grow LDs more slowly upon lipid loading? If not, this could further support the original claim of FA supply, and not TAG production.

Indeed, astrocytes treated with oleic acid continue to reduce the number of lipid droplets upon glutamate treatment. This new data can be found in **Fig 3D-F** of the revised manuscript. And lines 113-115 of the revised manuscript states “*glutamate reduced the number of lipid droplets in serum- free growth media (Fig. 3A,B). Lipid droplet size, however, remained unaffected (Fig. 3C). The same effect was observed when astrocytes were co-treated with oleic acid to promote lipid droplet formation (Fig. 3D-F).*” We speculate that because the effect is not dampened by the addition of exogenous lipids, it is possible that glutamate modulates LD biogenesis machinery directly. Therefore, we have also expanded our discussion section Lines 238-240 to state “*An additional mechanism that remains to be tested, is whether glutamate can also act directly on the lipid droplet biogenesis machinery. These two mechanisms could occur simultaneously to reduce the lipid droplet pool.*” This is a really interesting possibility. We thank the reviewer for this suggestion as we hope this study will stimulate future research on the topic.

*Minor - Upon a causal first read, the first two sentences of the abstract appeared to be conflicting. It eventually became clear, but it might be worth revising these for clarity.

We thank the reviewer for pointing this out. We have clarified the abstract of the revised manuscript to read “Astrocytes degrade fatty acids in response to glutamate while reducing the abundance of lipid droplets. But how glutamate regulates lipid droplets in astrocytes is unclear.”

Reviewer 2: SUMMARY OF THE ADVANCE MADE IN THIS PAPER AND ITS POTENTIAL SIGNIFICANCE TO THE FIELD

The manuscript by Rubio-Atonal et al. explores the effect of glutamate on fatty acid metabolism in astrocytes. The work is a follow-up on the very interesting and influential study by Ioannou et al. (2019), which showed that astrocytes take up FAs released from neurons and store them in lipid droplets (LDs) or consume via mitochondrial β -oxidation, protecting neurons from FA toxicity and oxidative stress during neuronal activity. As glutamate is the main excitatory neurotransmitter in the brain, studying its effect on astrocytes represents a simplified system to understand how astrocytes respond to neuronal activity. The authors show that glutamate activates the AMPK signaling pathway in primary hippocampal rat astrocytes, and that it is not taken up by the cells.

They observe a decrease in astrocyte LD size and a reduction in autophagy, suggesting that these processes are regulated by AMPK signaling, although they do not explain how. They propose that the reduced autophagy leads to a decrease in LDs, although again the causality and the temporal order of events are not established.

SUGGESTIONS TO AUTHORS

This manuscript addresses a very interesting question, pertinent to understanding the neuron/astrocyte interaction during sleep-wake cycle or in pathological states using a primary astrocyte cell model. This is surely a demanding experimental system that is not easy to manipulate. A number of processes are evaluated, often with a single experiment and a number of assumptions are not well justified, so the final model is not very convincing. A more critical assessment by the authors is needed. No such assessment is provided in the Discussion; the Discussion should be thoroughly revised.

We are pleased that the reviewer appreciates that this study addresses an interesting question and we thank the reviewer for their critical feedback. We have now revised the manuscript in many cases with the addition of new experiments that we believe further support our conclusions. We have also provided a more critical assessment in the discussion as recommended by the reviewer.

Specific comments:

1. The model used in the study are primary rat astrocytes cultured in serum-free medium, which is to maintain them in quiescent state. Why is it important to maintain quiescent state, and is this state relevant for the understanding of metabolic coupling between neurons and astrocytes?

We apologize that this was unclear. Our previous work showing neuron-to-astrocyte metabolic coupling used FBS-astrocytes. However, serum induces a reactive phenotype in astrocytes typically associated with disease (PMID: 28099414). We were interested in testing whether the same effect also occurs in an astrocytic model system that more closely resembles resting (non-reactive) astrocytes. Given that we found the same result in serum-free astrocytes, this suggests that this phenomenon is could be relevant beyond disease.

We have revised the manuscript as suggested by the reviewer to provide additional rationale for the use of serum-free cultures on lines 69-73 by stating “We previously found that glutamate modulates lipid physiology in astrocytes grown in serum (Ioannou, Jackson, et al., 2019a), which is known to induce a reactive phenotype typically associated with disease (Liddel et

al., 2017). To explore the effects of glutamate on lipids in resting-state astrocytes, we cultured primary rat astrocytes in a serum-free media that maintains astrocytes in a non-reactive state (Foo et al., 2011)."

We have also expanded our discussion section on lines 291-294 to state *"While our previous work shows that glutamate regulates lipid metabolism in serum-containing astrocyte cultures, the presence of serum induces a reactive phenotype typically associated with disease (Liddelow et al., 2017). This study using serum-free astrocytes suggests that these processes could also occur under more physiological states."*

We thank the reviewer for pointing this out. We think the new text will provide readers with a better understanding as to why using serum-free media further expands our understanding of these phenomena.

Furthermore, the authors use glutamate at 100 μM ; is this concentration physiologically relevant? More explanation regarding the chosen experimental conditions would be useful.

This is another a great suggestion. The extracellular concentration of glutamate in the brain varies dramatically. This particular concentration was chosen to mirror that used to stimulate lipid release from neurons and reduce astrocytic lipid droplets in our previous study.

We have now revised the results section to provide rationale for choosing this concentration. Lines 80-84 we state *"Given that extracellular glutamate concentration in the brain varies by several orders of magnitude depending on the compartment and/or biological context (Dzubay & Jahr, 1999; Kalivas, 2011), 100 μM glutamate was used as this concentration was previously shown to increase lipid transport from neurons to astrocytes and reduce astrocyte lipid droplets (Ioannou, Jackson, et al., 2019)."*

We have also expanded our discussion on the topic. This new text found on lines 301-307 states *"A further consideration of this work is the concentration of glutamate used to elicit these effects. This study used 100 μM glutamate as was previously shown to regulate astrocytic lipid droplets (Ioannou, Jackson, et al., 2019a). But the extracellular concentration of glutamate varies dramatically in the brain; typically, from 1-30 μM and climbing to 1 mM in the synapse following an action potential (Dzubay & Jahr, 1999; Kalivas, 2011). Whether these different pools of glutamate are sufficient to regulate lipid droplets and oxidative stress in astrocytes during health or whether this metabolic coupling is more important during periods of excitotoxicity has yet to be determined."*

2. In Fig 2, mitochondrial activity is assessed using Seahorse assays, and in combination with different inhibitors to determine which fuel is used by the mitochondria (pyruvate, glutamate/glutamine or fatty acids). No reference is given for these inhibitors, and no control is provided to assess their activity or specificity, so it's not possible to judge how reliable are these results. Only a single inhibitor per pathway is used.

The Seahorse Assays Fuel Flex assay is purchased as a kit that come with those specific drugs and concentration of drugs was used according to the manufacturer's protocol. As suggested by the reviewer, we have now included references for the Seahorse Assays that utilize the same drugs concentrations as we have in our study. For brevity, we have only included references to studies using astrocytes. This new text can be found in the methods section on Lines 462-464 of the revised manuscript *"...were performed according to the manufacturer's protocol using an Agilent Seahorse xFe96 Metabolic Analyzer and as previously described (Mi et al., 2023; Qi et al., 2021; Weightman Potter et al., 2019)."*

Additional support for these findings comes from the literature and our own validation experiments. The finding that glutamate/glutamine oxidation increases when glutamate is added to astrocytes has been reported extensively (reviewed nicely in PMID: 24379804). The addition of glutamate to astrocytes has also been shown previously to decrease glucose oxidation by 75% (PMID: 15973351). Lastly, the drug used to assess fatty acid oxidative, etomoxir, was validated in the preceding figure panels. This drug effectively reduced oxidative metabolism in the presence of palmitate (Fig 1E-F) but not in the absence of exogenous lipids

(Fig 2A-B). Therefore, we believe our experiments reliably show the fuel pathways used by astrocytes in the presence of glutamate,

In general, the study relies heavily on the use of inhibitors, and we have to trust that the inhibitors function as stated. TFB-TBOA is the most problematic because it has no effect and no information is given...

We agree that TFB-TBOA is an important drug used in the study and that it is especially important to validate its actions given that it has no effect on LDs or ROS. For this reason, we confirmed its ability to block entry of glutamate into the cell. This data was provided in Fig 6A of the original manuscript and can now be found in Fig 7A of the revised manuscript.

...compound C is inhibitory, but is it specific? At the least, more information is needed, whereas alternative approaches and controls would strengthen the conclusions.

We agree with the reviewer that compound C may have off-target effects in addition to its ability to inactivate AMPK. Previously we found that inactivating AMPK with compound C increases the number of lipid droplets. We performed new experiments and found that conversely, activation of AMPK with two different drugs can reduce the number of lipid droplets in astrocytes. This new data can be found in Fig 6I-K of the revised manuscript and lines 187-190 “Conversely, AMPK activators, AICAR and Metformin, reduce the number of lipid droplets in the absence of glutamate (Fig. 6I-K). These data indicate that glutamate activates AMPK in astrocytes which can contribute to the reduction in lipid droplets upon glutamate treatment.” We believe these alternate approaches strengthen the conclusion that AMPK sufficiently regulates lipid droplets in astrocytes.

3. In all figures, the authors state the number of independent experiments and the number of replicates per experiment, but it is not stated how many cells were analyzed in each replicate, and therefore what was the n used for statistical analyses. This needs to be clarified in all figure legends and the statistical tests used should be better justified.

All statistical analyses were performed on independent experiments as required by the Journal of Cell Science. However, we apologize for the confusion surrounding the number of technical replicates as they were buried in the methods and disconnected from the relevant number of independent experiments. We have revised the manuscript to indicate the number of technical replicates directly in the figure legend. To avoid further confusion, we now display the figures as Superplots and have modified the methods section to indicate on lines 502-506 that “All graphs are depicted as SuperPlots where independent replicates are shown in large shapes, and the technical replicants shown as small grey shapes (Lord et al., 2020). The independent replicate values were calculated from the mean of technical replicates. Statistics analyses were performed on the independent replicates for each experiment.” We appreciate the reviewer highlighting this point, as we believe the revised manuscript will now be much easier for our readers to follow.

4. The authors show that glutamate decreases LD number, but not LD size. However, the effect on LD number varies between figures: 3-fold decrease in Fig.3 ($p < 0.001$) vs 2-fold in Fig. 4 ($p < 0.05$) vs. 1.5-fold in Fig. 7E ($p < 0.05$). As they are performed now, the statistical tests don't seem to be vary reliable.

The variance in LD number is typical of working with batches of primary cells. Moreover, we do not believe it is meaningful to compare p-values across experiments where different statistical tests are used, for example t-tests (Fig 3) versus ANOVA with corrections for multiple comparisons (Figs 4 and 7) in the original manuscript. Instead, we believe that since the same result (glutamate reduces LD numbers but not size) has been repeated 5 times throughout the study (by 4 different experimenters) argues how reliable this finding truly is.

- The authors argue that this decrease is not due to an increase in lipolysis because LD size is not affected, citing a study in liver cells; what is the evidence that liver cells are a relevant comparison?

This is a good point as astrocytes may behave differently than liver cells. We have now cited an additional paper showing that TG lipase DDHD2 modulates LD size in the brain. On line 126-127, we state “Stimulating lipolysis shrinks lipid droplets, whereas inhibiting lipolysis causes lipid droplets to expand in size (Inloes et al., 2014; Schott et al., 2019).”

We also performed new experiments in the presence of DGAT1/2 inhibitors and found that glutamate no longer affects LD numbers. This new data can be found in Fig 4G-I of the revised manuscript and referred to on Lines 133-137 we state “Finally, we tested whether glutamate continues to reduce lipid droplets in the absence of triglyceride synthesis by inhibiting diacylglycerol acyltransferase (DGAT) 1 and 2. We reasoned that if lipolysis was increased, then glutamate would further reduce lipid droplets in the absence of DGAT1/2 activity. However, glutamate had no additional effect on lipid droplets in the presence of DGAT-1 and 2 inhibitors (Fig. 4G-I).” We infer that since that glutamate no longer affects the number

of LDs when triglyceride synthesis is blocked, that glutamate is more likely acting on the formation of LDs rather than the degradation.

- The other argument is that free fatty acids are not increased, but there actually is some increase in free fatty acids (fig. 4D), except it is not assessed whether this increase is statistically significant.

We apologize for this oversight. The total levels of free fatty acids were assessed and not statistically significant. We have now indicated this directly on the figure panel (Fig 4D) in the revised manuscript to avoid further confusion. We thank the reviewer for pointing this out. We also explore individual lipid species as described in more detail below confirming there are no alterations in free fatty acids.

There is a large variability between experiments; plotting data in super plots would allow visual comparison between individual experiments.

Indeed, large variability is typical for these types of experiments. We have modified the figures to display as superplots as suggested by the reviewer. This is a great suggestion as it allows the readers to better appreciate the variability of the data!

Taking into account the variability between experiments, might there be a difference in each experiment between +/- glutamate conditions? And how much of an increase in free FAs can be expected in these cells if lipolysis is induced?

The reviewer raises a valid point. We performed high sensitivity lipidomics analysis of free FAs from 5 independent experiments. In other cell types, increased free FAs from lipolysis can be easily detected using less sensitive methods such as TLC (PMID: 25752962). Therefore, our methods should detect changes in FAs induced by lipolysis. However, it is possible that differences in certain FAs might be masked by high variability as the reviewer states. We have now included a heatmap showing the fold-change of all FA species measured from each independent experiment. This new data, found in Fig 4F of the revised manuscript, confirms that there are no changes in any FA species with glutamate treatment, and further strengthens the conclusions of the paper

5. Mitochondrial volume is increased (Fig. 5); what is the connection with lipid droplets and oxidative metabolism? Contradicting statements seem to be given in different parts of the manuscript.

When ROS is high, two things can happen 1. mitochondria become fragmented which decreases their capacity for oxidative metabolism. 2. mitochondria are degraded by mitophagy (further reducing oxidative metabolism) and the resulting lipids from mitochondrial membranes are converted to TGs and stored in lipid droplets. When astrocytes are treated with glutamate, ROS is reduced which protects mitochondria. More fused mitochondria are more efficient at oxidative metabolism while decreased mitophagy means less lipids in need of storage in lipid droplets. We hope this clarifies any confusion regarding the statements made throughout the manuscript.

Second decision letter

MS ID#: jcs.263983R1

MS Title: Glutamate decreases oxidative stress and lipid droplet formation in astrocytes

Authors: Luis F Rubio-Atonal; Jinlan Chang; Julie Jacquemyn; Isha Ralhan; Iset Ilarraza; Maria Ioannou

Article Type: Research Article

Dear Dr Ioannou,

I am happy to tell you that your manuscript has been accepted for publication in Journal of Cell Science, pending standard publication integrity checks.